# Self-supervised Amodal Video Object Segmentation

**Jian Yao**[1]*    **Yuxin Hong**[2]*    **Chiyu Wang**[3]*    **Tianjun Xiao**[4]†    **Tong He**[4]
**Francesco Locatello**[4]    **David Wipf**[4]    **Yanwei Fu**[2]†    **Zheng Zhang**[4]

[1] School of Management, Fudan University
[2] School of Data Science, Fudan University
[3] University of California, Berkeley
[4] Amazon Web Services

{jianyao20, yxhong20, yanweifu}@fudan.edu.cn, wcy_james@berkeley.edu
{tianjux, htong, locatelf, daviwipf, zhaz}@amazon.com

## Abstract

Amodal perception requires inferring the full shape of an object that is partially occluded. This task is particularly challenging on two levels: (1) it requires more information than what is contained in the instant retina or imaging sensor, (2) it is difficult to obtain enough well-annotated amodal labels for supervision. To this end, this paper develops a new framework of Self-supervised amodal Video object segmentation (SaVos). Our method efficiently leverages the visual information of video temporal sequences to infer the amodal mask of objects. The key intuition is that the occluded part of an object can be explained away if that part is visible in other frames, possibly deformed as long as the deformation can be reasonably learned. Accordingly, we derive a novel self-supervised learning paradigm that efficiently utilizes the visible object parts as the supervision to guide the training on videos. In addition to learning type prior to complete masks for known types, SaVos also learns the spatiotemporal prior, which is also useful for the amodal task and could generalize to unseen types. The proposed framework achieves the state-of-the-art performance on the synthetic amodal segmentation benchmark FISHBOWL and the real world benchmark KINS-Video-Car. Further, it lends itself well to being transferred to novel distributions using test-time adaptation, outperforming existing models even after the transfer to a new distribution. The experiment code is available at https://github.com/amazon-science/self-supervised-amodal-video-object-segmentation.

## 1 Introduction

Cognitive scientists have found human vision system contains several hierarchies. Visual perception [27] first carves a scene at its physical joints, decomposing it into initial object representation by grouping and simple completion. At this point, the representation is tethered into the retina sensor [13]. Then, correspondence or motion on the temporal dimension is built to form the object representation that is untethered from the retinal reference frame through operations like spatiotemporal aggregation, tracking, inference and prediction [6]. The more stable untethered representation is ready to be raised from the perception system to the cognitive system for higher level action and symbolic cognition [23]. Machine learning, especially with artificial neural networks, has progressed tremendously on tethered vision tasks like detection and modal segmentation. The natural next step is to go the higher rung of the ladder by tackling untethered vision.

---

*Work completed during internship at AWS Shanghai AI Labs.
†Correspondence authors are Tianjun Xiao and Yanwei Fu.

36th Conference on Neural Information Processing Systems (NeurIPS 2022).

This paper studies the task of amodal segmentation which aims at inferring the whole shape of the object on both visible and occluded parts. It has critical applications on robot manipulation and autonomous driving [24]. Conceptually, this task is on the bridge between tethered and untethered representations. Amodal segmentation requires prior knowledge. One option that has been explored in literature is using the tethered representation and prior knowledge about object type to get amodal mask. Alternatively, we can get amodal masks using the untethered representation by building dense object motion across frames to explain away occlusion, which is referred as spatiotemporal prior. We prefer to explore more on the second one since the dependence on type prior makes the first method hard to generalize, considering the frequency distribution of visual categories in daily life is long-tailed.

Following this direction, we propose a Self-supervised amodal Video object segmentation (SaVos) pipeline which simultaneously models amodal mask and the dense object motion on the amodal mask. Unlike traditional optical flow or correspondence networks, our approach does not require explicit visual correspondence across pixels, which would be impossible due to occlusions. Instead, modeling motion using temporal information allows us to complete dense amodal motion predictions.

The architecture is built for spatiotemporal modeling, which has better generalization performance than using type priors. Despite that, we show that SaVos automatically figures its way to learn type prior as well, as learning types can help the encoder-decoder-style architecture make prediction. This makes generalization to distribution shifts remain challenging, for example, to unseen types of objects. To address this issue, we need to suppress the type prior and amplify spatiotemporal prior to make predictions. This is achieved by combining SaVos with test-time adaptation. Critically, we found that our model is "adaptation-friendly" as it can naturally be improved with test-time adaptation techniques without any change on the self-supervised loss, achieving a significant boost in generalization performance.

We make several contributions in this paper:

(1) We propose a Self-supervised amodal Video object segmentation (SaVos) training pipeline built upon the intuition that the occluded part of an object can be explained away if that part is visible in other frames (Figure 1), possibly deformed as long as the deformation can be reasonably learned. The pipeline turns visible masks in other frames to amodal self-supervision signals.

(2) The proposed approach simultaneously models the amodal mask and the dense amodal object motion. The dense amodal object motion builds the bridge between different frames to achieve the transition from visible masks to amodal supervision. To address the challenge of predicting motion on the occluded area, we propose a novel architecture design that takes the advantage of the inductive bias from the spatiotemporal modeling and the common-fate principle of Gestalt Psychology [39]. The proposed method shows the state-of-the-art amodal segmentation performance in self-supervised setting on several simulation and real-world benchmarks.

(3) The proposed SaVos model shows strong generalization performance on drastic distribution shifts between training and test data after combining with one-shot test-time adaptation. We empirically demonstrate that, by applying test-time adaptation without any change on the loss, SaVos trained on synthetic fish dataset can even outperform a competitor that is well learned on the target real-world driving car dataset. Interestingly, applying test-time adaptation on an image-level baseline model doesn't bring the same improvement as observed on SaVos. This provides an unique perspective on comparing different models by checking how effective can test-time adaptation work on them.

## 2   Related works

**Untethered vision and amodal segmentation**. Human vision forms a hierarchy by grouping retina signals into initial object concept; and the representation will untether from the immediate retina sensor input grouping the spatiotemporally disjoint pieces. Such untethered representation has been studied in various topics [23, 21, 28, 26, 34, 22]. Particularly, Amodal segmentation [46] is a task inferring shape of the object on both visible and occluded part. There are various image amodal datasets such as COCOA [46] and KINS [24], and video amodal dataset – SAIL-VOS [11] created by the GTA game engine. Unfortunately, SAIL-VOS has frequent camera view switches, not the ideal testbed to apply video tracking or motion. Several efforts are made towards amodal segmentation on these datasets [46, 24, 7, 45, 41, 44, 33, 17, 43, 30, 20]. Generally speaking, most of the methods are on image level and they model type priors with shape statistics, as such it is challenging to extend

their models to open-world applications where object category distributions are long-tail. Amodal segmentation is also related to structured generative model [47, 16, 29, 18]. These models attempt to maximize the likelihood of the whole video sequences during training so as to learn a more consistent object representation and the amodal representation. However, the major tasks for those models are object discovery and presentation and they are tested on simpler datasets; self-supervised object discovery in real-world complex scenes like the driving scene in [9] remains too challenging for these methods. Without object discovered, no proper amodal prediction can be expected.

**Dense correspondence and motion** Our goal is to achieve amodal using untethered process, which requires object motion signals. There have been studies [8, 2] on correspondence and motion before deep learning time. FlowNet [5] and its follow-up work FlowNet2 [14] train deep networks in a supervised way using simulation videos. Truong et al.[34] proposes GLU-Net, a global-local universal network for dense correspondences. However, motion on the occlusion area cannot be estimated with those methods. Occlusion and correspondence estimation depend on each other and it is a typical chicken-and-egg problem [15]. We need to model additional priors.

**Video inpainting** A related but different task is video inpainting. Existing video inpainting methods fill the spatio-temporal holes by encourage the spatial and temporal coherence and smoothness [40, 10, 12], rather than particularly inferring the occluded objects. The object-level knowledge was not explicitly leveraged to inform the model learning. Recently, Ke et al.[17] learns object completion by contributing the large-scale dataset Youtube-VOI, where occlusion masks are generated using high-fidelity simulation to provide training signal. Nevertheless, there is still the *reality gap* between synthetic occlusions and the amodal masks in the real-world . Accordingly, our model is designed to learn the amodal supervision signal in the easily accessible raw videos if spatiotemporal information is properly mined.

**Domain generalization and test-time adaptation** Transfer learning [4] and domain adaptation [25] are general approaches for improving the performance of predictive models when training and test data come from different distributions. Sun et al.[31] proposes Test-time Training. It is different from finetuning where some labeled data are available in test domain, and different from domain adaptation where there is access to both train and test samples. They design a self-supervised loss to train together with the supervised loss and in test time, apply the self-supervised loss on the test sample. Wang et al.[37] proposes a fully test time adaptation by test entropy minimization. A related topic is Deep Image Prior [36] and Deep Video Prior [19], they directly optimize on test samples without training on training set. Our model is self-supervised thus naturally fit into test-time adaptation framework. We will see how it works for our method on challenging adaptation scenarios.

## 3 Method

**Notations** . Given the input video $\{\mathbf{I}_t\}_{t=1}^{T}$ of $T$ frames with $K$ objects, the task is to generate the amodal "binary" segmentation mask sequences $\mathbf{M} = \{M_t^k\}$ for each object in every frame. On the raw frames, we obtain the image patch $I_t^k$ and visible modal segmentation mask $\mathbf{V} = \{V_t^k\}$. Further, we also obtain the optical flow $\Delta V_t = \{\Delta V_{x,t}^k, \Delta V_{y,t}^k\}$ such that $I_{t+1}[x + \Delta V_{x,t}[x,y], y + \Delta V_{y,t}[x,y]] \approx I_t[x,y]$. These information can be retrieved from human annotation or extracted with off-the-shelf models, and we use them as given input to our model.

### 3.1 Overview of Our SaVos Learning Problem

The key insight of the amodal segmentation task is to maximally exploit and explore visual prior patterns to *explain away* the occluded object parts [35]. Such prior patterns include, but are not be limited to (1) **type prior**: the statistics of images, or the shape of certain types of objects; and (2) **spatiotemporal prior**: the current-occluded part of an object might be visible in the other frames, as illustrated in Figure 1. Under a self-supervised setting, we exploit temporal correlation among frames relying exclusively on the data itself.

Specifically, SaVos generates training supervision signals by investigating the relations between the amodal masks and visible masks on neighboring frames. The key assumption is "some part is occluded at some time, but not all parts all the time", and that deformation of the past visible parts can be approximately learned. I.e. gleaning visible parts over enough frames produces enough evidences to complete an object. The inductive bias we can and must leverage is spatiotemporal continuity.

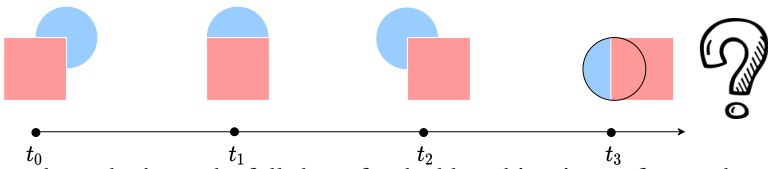

Figure 1: Though we don't see the full shape for the blue object in any frames, human can tell the full shape of that object at frame $t_3$. We just need to stitch seen parts in different frames.

We note that parts remain occluded all the time can not be recovered unless there are other priors such as classes and types, which is also learnable in our model design. Figure 2 a) shows the training pipeline.

**Prediction of amodal mask and amodal motion** At frame $t$ for object $k$, SaVos predicts the amodal mask $\tilde{M}_t^k$ with

$$\tilde{M}_t^k, \Delta \tilde{M}_t^k = F_\theta(I_{\leq t}^k, V_{\leq t}^k, \Delta V_{\leq t}^k) \tag{1}$$

where $F_\theta$ is a learnable module paramterized by $\theta$ with more detailed introduction in Section 3.3, and $\Delta \tilde{M}_t^k$ is the dense motion on the amodal mask. Ideally $F_\theta$ is able to aggregate available visual, semantic and motion information from all the currently available frames. The intuition here is that seeing various parts of an object (i.e. $\mathbf{V}$) and their motions ($\mathbf{\Delta V}$) is enough to reconstruct $\mathbf{M}$. However, leaving as is, there can be an infinite number of explain-away solutions.

The first obvious training signal is to check the consistency between $\tilde{M}_t^k$ and $V_t^k$. The signal is weak since the model can learn a simple copy function. A stronger signal is to assume a generative component to predict the amodal mask in the *next* time frame and use visible $V_{t+1}^k$ in addition. Since transformation is already predicted by $\Delta \tilde{M}_t^k$, we can obtain an estimation of the amodal mask at frame $t+1$ by a warping function, i.e. $\hat{M}_{t+1}^k = \text{Warp}(\tilde{M}_t^k, \Delta \tilde{M}_t^k)$. $\hat{M}_{t+1}^k$ satisfies

$$\hat{M}_{t+1}^k[x + \Delta \tilde{M}_{x,t}^k[x,y], y + \Delta \tilde{M}_{y,t}^k[x,y]] \equiv \tilde{M}_t^k[x,y]. \tag{2}$$

Now we compute the distance between $\hat{M}_{t+1}^k$ and $V_{t+1}^k$ with the assumption that $V_{t+1}^k$ might includes parts occluded in $V_t^k$, and define the first training loss as

$$\mathcal{L}_M = \sum_{k=1}^K \sum_{t=1}^{T-1} d_M\left(\hat{M}_{t+1}^k, V_{t+1}^k\right) \tag{3}$$

where

$$d_M\left(\hat{M}_{t+1}^k, V_{t+1}^k\right) = W_{t+1}^k \odot \text{BCE}\left(\hat{M}_{t+1}^k, V_{t+1}^k\right) \tag{4}$$

with BCE being the vanilla form of binary cross entropy loss function, and $W_{t+1}^k$ being a weight matrix that masks out the area occluded at $t+1$ for object $k$. Concretely,

$$W_{t+1}^k = \left(\mathbf{1} - \sum_{i=1}^K V_{t+1}^i\right) + V_{t+1}^k \tag{5}$$

where $\mathbf{1}$ is a all-one matrix with the same shape as the mask tensor. With this mask, $\mathcal{L}_M$ only generates supervision signal on the visible parts and background between different frames, with the occluded part being masked out from providing any feedback. Our prediction can be *under*-complete if only $\mathcal{L}_M$ is applied, since all that the model is forced to learn is the visible masks one frame later.

**Amodal consistency loss** The temporal consistency loss $\mathcal{L}_C$, assuming some distance measure function $d(\cdot, \cdot)$ is straightforward in its form:

$$\mathcal{L}_C = \sum_{k=1}^K \sum_{t=2}^T d_C\left(\tilde{M}_t^k, \hat{M}_t^k\right) \tag{6}$$

The intuition is that the amodal mask prediction $\tilde{M}_t^k$ at $t$ should be consistent with the estimation $\hat{M}_t^k$ warped from $t-1$. $\tilde{M}_t^k$ includes new evidence $(V_t^k, \Delta V_t^k)$ that is not available when computing

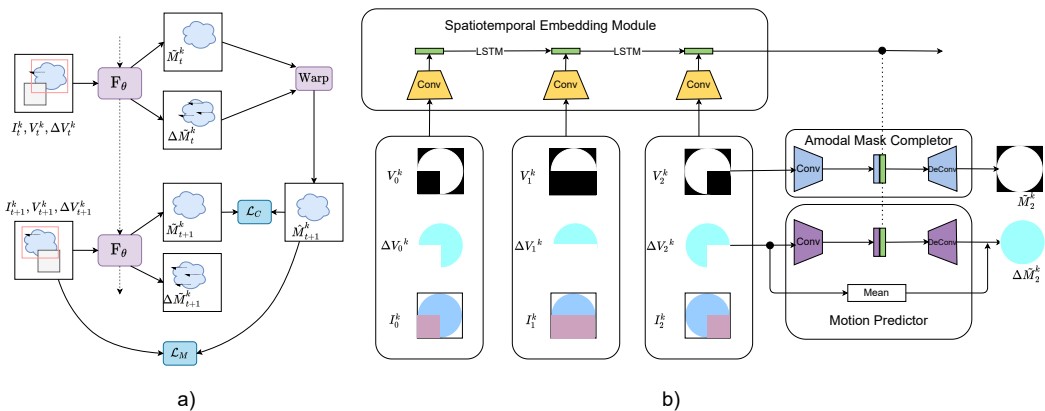

Figure 2: a) SaVos Training Pipeline. b) SaVos Architecture

$\hat{M}_t^k$, forcing the generative component in estimating $\hat{M}_t^k$ to catch up. This loss links the supervision signals in all frames to guide the prediction in each frame.

Consider an object that is made up by two parts, each is visible in only one of the two adjacent frames, a good estimation is encouraged to include both parts under this loss. In this work, we use Diff_IoU as the metric $d_C$, introduced in [3], since it's symmetric to inputs. Note that this loss has a similar form of temporal cycle-consistency [38].

Combining Equation 4 and Equation 6, the final loss is:

$$\mathcal{L} = \lambda_1 \cdot \mathcal{L}_M + \lambda_2 \cdot \mathcal{L}_C \tag{7}$$

**Analysis on necessity and sufficiency of** $\mathcal{L}$ Define $\mathcal{L}_M^k = \sum_{t=1}^{T-1} d_M \left( \hat{M}_{t+1}^k, V_{t+1}^k \right)$ and $\mathcal{L}_C^k = \sum_{t=2}^{T} d_C \left( \tilde{M}_t^k, \hat{M}_t^k \right)$. It is easy to show that $\mathcal{L}_M^k = \mathcal{L}_C^k = 0$ is necessary for $\tilde{M}_t^k = \hat{M}_t^k = M_t^k$. Specifically, for any $t$ and $k$, we have $d_M \left( \hat{M}_t^k, V_t^k \right) = d_C \left( \tilde{M}_t^k, \hat{M}_t^k \right) = 0$ if $\tilde{M}_t^k = \hat{M}_t^k = M_t^k$, and that directly leads to $\mathcal{L}_M^k = \mathcal{L}_C^k = 0$. We further analyzes the sufficiency of $\mathcal{L}_M^k = \mathcal{L}_C^k = 0$. The theorem statement and its proof can be found in supplementary material.

Some pixels of an object might never be visible in any frames. For example, parked cars that line up along the roadside with corners invisible, e.g. the target car in Figure 3. We emphasize that our model can still align the correct amodal segmentation with the global optima of the proposed loss. Consequently, on those cases, our method can still work at least as good as image-level methods since it is also able to capture type prior for known types with the architecture design in Section 3.3. The encoder-decoder architecture contains an information bottleneck, which makes it easy to squeeze out type information since it's concise and beneficial to amodal prediction. In addition, the fewer pixels remain invisible all the way, the more spatiotemporal information our model can leverage to improve its performance.

**Bi-directional prediction** SaVos as described so far suffers from cold start problem, i.e. the first few frames may not be informative enough. We solve this by simply predicting backwards in time. To merge the prediction from both directions, we add an alpha channel prediction together with the amodal mask and let the model decide which side to trust more:

$$\hat{M}_t^k = \overrightarrow{\hat{\alpha}_t^k} \odot \overrightarrow{\hat{M}_t^k} + \overleftarrow{\hat{\alpha}_t^k} \odot \overleftarrow{\hat{M}_t^k} \tag{8}$$

where $\overrightarrow{\hat{\alpha}_t^k}$ and $\overrightarrow{\hat{\alpha}_t^k}$ are the alpha channel from each direction normalized with each other using Softmax function. $\overrightarrow{\hat{M}_t^k}$ and $\overleftarrow{\hat{M}_t^k}$ are the predicted amodal mask from each direction.

### 3.2 Test-time Adaptation for SaVos

SaVos models spatiotemporal prior and as such should be robust against data distribution shift. However, certain components (especially the generative part) are sensitive to samples in the training data. Type prior can be implicitly learned during training, which the model falls back on and struggles

to "synthesize" novel masks. In this work we adopt one-shot test-time adaptation as stated in [32]. We don't expect new knowledge to be learned on single sample, but to suppress unnecessary type prior and rely only on learned spatiotemopral prior. The advantage is that a base model can be reused to tackle new data distribution which is not expected to be part of the long term sample repository.

Since training is on each single test sample independently, no change for the testing environment is required, only the inference time will increase. In [32], a test-time self-supervised loss is used to help the model adapt to the test data distribution. Theoretically, if the gradient of the test-time loss is on the same direction of the main training-time loss, the model performance on the test domain will be improved. As our SaVos model is self-supervisedly learned, the test-time adaptation naturally apply. In practice, we optimize the loss on Equation 7 on a test video *without any change*.

In experiments, test-time adaptation indeed helps on challenging distribution shifts test scenarios. We will have more detailed analysis on the learning dynamics and efficiency on the test-time adaptation for SaVos in the Experiment Section 4 comparing with the baseline.

## 3.3 Architecture

As depicted in Figure 2 b), the overall architecture has three components: 1) the spatiaotemporal embedding that summarizes object-level signals into a hidden representation $h_t^k$, 2) the amodal mask completor that takes $h_t^k$ and $V_t^k$ to output the estimated amodal mask $\tilde{M}_t^k$, and 3) the estimated amodal mask motion $\Delta \tilde{M}_t^k$. The generator function Warp itself does not contain any parameters.

**Spatiotemporal embedding module** This module extracts features from video frames, aligns and aggregates the feature across frames. The encoder (Enc) takes the concatenation of raw image patches, optical flow patches and visible masks as input. Then a recurrent architecture (Seq) aggregates information through temporal dimension.

$$f_t^k = \text{Enc}\left(I_t^k, \Delta V_t^k, V_t^k\right) \tag{9}$$

$$h_t^k = \text{Seq}\left(h_{t-1}^k, f_t^k\right) \tag{10}$$

$h_t^k$ is the spatiotemporal embedding for object $k$ at frame $t$. Here, we implement the encoder Enc and Seq with CNNs and LSTMs, respectively. This module also learns reasonable deformation over time.

**Amodal mask completor** Amodal mask completor is an Encoder-Decoder architecture with an information bottleneck. The CNN encoder takes the visible mask $V_t^k$ and concatenate the output with $h_t^k$, then uses several de-convolutional layers DeConv to produce the full mask prediction:

$$\tilde{M}_t^k = \text{DeConv}_a\left(\left[h_t^k, \text{CNN}_a\left(V_t^k\right)\right]\right) \tag{11}$$

where $a$ is the parameters of the CNN and DeConv above.

**Motion predictor** Computing $\Delta \tilde{M}_t^k$ uses the same general encoder-decoder architecture as the amodal mask completor except it takes $\Delta V_t^k$ instead of $V_t^k$ as input. In addition, the computation takes the form of residual prediction, using the mean of visible mask motion signal $\Delta \bar{V}_t^k$ as the base to correct. This inductive bias reflect the common-fate principle of Gestalt Psychology[39].

$$\Delta \tilde{M}_t^k = \Delta \bar{V}_t^k + \text{DeConv}_c\left(\left[h_t^k, CNN_c\left(\Delta V_t^k\right)\right]\right) \tag{12}$$

where $c$ is the parameters of the CNN and DeConv above.

## 4  Experiments

We evaluate the proposed pipeline on both close-world setting with no distribution shifts between training set and test set, as well as the setting has distribution shifts with new object types.

**Chewing Gum Dataset** Chewing Gum is a synthetic dataset consists of random generated polygons (that look like chewing gum). Each polygon has a random number of nodes ranging from 7 to 12 and the nodes randomly scattered on a circle. Object are occluding each other and have relative movement. The occluded object never shows its full shape. But each part is shown in at least one frame. Because of the randomness in generation, the shape prior for a certain type will not work.

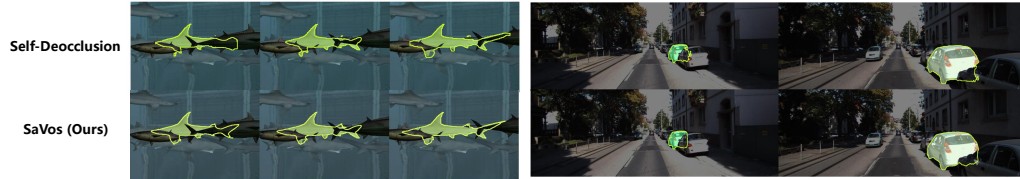

Figure 3: Qualitative comparison between the image-level baseline Self-Deocclusion [44] and our SaVos model. Our model produces more consistent predictions across frames and the performance is better when the occlusion rate is high. This is expected considering the consistency loss and the spatiotemporal architecture SaVos has.

**FISHBOWL Dataset** This dataset [33] consists of 10,000 training and 1,000 validation and test videos recorded from a publicly available WebGL demo of an aquarium [1], each with 128 frames with resolution at 480×320. It is positioned between simplistic toy scenarios and real world data.

**KINS-Video-Car Dataset** KINS is an image-level amodal dataset labeled from the city driving dataset KITTI [9]. In order to have SaVos work with KINS, we match images in KINS to its original video frame in KITTI. Since only training videos are available online, we re-split the original KITTI training set into three subsets for training, validation and test. We use PointTrack [42] to extract visible masks and object tracks to drive our video-based algorithm. We only run the algorithm for the `Car` category and mark this modified KINS dataset as `KINS-Video-Car`.

**Metrics and Settings**. The metric to evaluate amodal segmentation is mean-IoU as in [44, 33, 24]. Specifically, we compute mean-IoU against the groundtruth full mask as well as only the occluded sub-area, in order to evaluate the overall and focused performance. Occluded mean-IoU is usually a better indicator for amodal segmentation. We use groundtruth visible mask and tracking as inputs for FISHBOWL and Chewing Gum, and pre-compute visible mask and tracking from PointTrack[42] model for KINS-Video-Car. Note that self-supervision in this work is only for the amodal mask completion. On FISHBOWL, we only compute mean-IoU for objects with the occlusion rate from 10% to 70%. On KINS-Video-Car, we match the tracked objects and the annotated ones from KINS and only compute mean-IoU on the paired ones. All self-supervised models share the same input, while the supervised baseline is trained on samples with labels. For test-time adaptation, one test video is given and adapted separately. We run repeating experiments on our own method but not all baselines since the training is costly. The performance difference between runs is around 0.02 on the occluded mean-IoU. Pariculalry, we further propose two new settings to evaluate the performance on distribution shifts. In the first setting, we train a model on four type of fishes in FISHBOWL and test on the rest type. In the second one, we train a model on FISHBOWL and evaluate on KINS-Video-Car.

**Competitors**. We use a simple heuristic method that just completes the object into a convex shape, a state-of-the-art image-level self-supervised model Self-Deocclusion [44], and supervised oracles: a recent state-of-the-art supervised method VRSP-Net [41] or U-Net [28], depending on the availability of the codebook for VRSP-Net.

### 4.1   Experiment Results on Test Set with No Distribution Shifts

**Performance on FISHBOWL and KINS-Video-Car** Table 1 compares SaVos with baselines. Qualitatively, Figure 3 compares predictions between SaVos and Self-Deocclusion. Note that KINS-Video-Car poses several challenges: the model-inferred object visible mask and tracking are inevitably inaccurate; videos with camera motion bring complex temporal motion signals like zoom-in/out, lens distortion and change of view point. However, our model still works on this challenging scenario. Also, some pixels for target car in the video of Figure 3 b) are never visible in any frames. This is a representative case for the parked cars that line up along roadside with corners always invisible. Our model still produce complete amodal mask, which can not be achieved if only spatiotemporal prior is learnt. This is an indicator that Savos also learns type prior during training.

**Performance on Chewing Gum dataset** Chewing Gum is `classless` as every two objects are different. As shown in Table 1, all image-level models fail to predict the occluded part. Since there is no type prior, no image-level model, including the supervised one, can do much better than

Table 1: Mean-IoU on FISHBOWL, KINS-Video-Car and Chewing Gum datasets. For the supervised oracle results, we use VRSP-Net[41] for KINS-Video-Car, and U-Net[28] for FISHBOWL and Chewing Gum since the codebook for VRSP-Net is not available for these two datasets.

| Method | Supervised | FISHBOWL | | KINS-Video-Car | | Chewing Gum | |
| | | Full | Occluded | Full | Occluded | Full | Occluded |
|---|---|---|---|---|---|---|---|
| Convex | ✗ | 0.7761 | 0.4638 | 0.7862 | 0.0829 | 0.9264 | 0.3182 |
| Self-Deocclusion [44] | ✗ | 0.8704 | 0.6502 | 0.8158 | 0.1790 | 0.9624 | 0.3307 |
| SaVos (Ours) | ✗ | **0.8863** | **0.7155** | **0.8258** | **0.3132** | **0.9746** | **0.8046** |
| Supervised Oracle | ✓ | 0.9162 | 0.7500 | 0.8551 | 0.4883 | 0.9613 | 0.3321 |

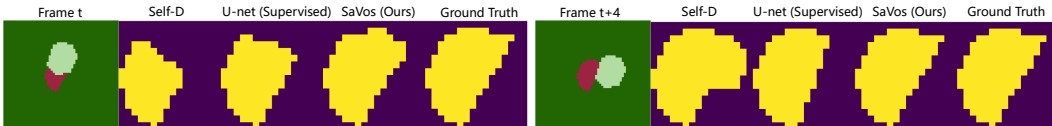

Figure 4: Amodal prediction on a Chewing Gum video on different frames. SaVos predicts consistent masks while for the image-based method the prediction varies even for the supervised method.

completing the object to its convex hull. SaVos outperforms the rest with a significant advantage. Note that the occluded part only occupies a small portion of the entire full mask, thus mean-IoU on the occluded part is a better indicator on the model performance. Visualization is shown in Figure 4.

## 4.2 Experiment Results on Test Set with Distribution Shifts

It's a major challenge for machine learning models to generalize under distribution shifts and test videos may contain objects from unseen types. Amodal methods depending only on type prior suffers from these challenges, while a model that considers spatiotemporal prior can work better. When the model learns both type and spatiotemporal prior, test-time adaptation strategy can pick-up spatiotemporal prior to achieve good generalization performance. We verify this statement in the following experiments.

**Test-time adaptation performance on FISHBOWL dataset with unseen fishes** Trained on 4 types of fish, SaVos tries to fit a new type of fish out of the ones it has already learned (Figure 5). This is an indicator SaVos also learns type prior. Savos with test-time adaptation produces competitive result compared with a model trained with all types of fishes (Table 2). The image-level baseline model, although also test-time trained, doesn't show the same amount of improvement. This demonstrates that SaVos has picked up spatiotemporal prior as designed.

**A more challenging scenario: adapting the model trained on FISHBOWL to KINS-Video-Car dataset**. To further check the ability of adaptation on visually complex data, we use a model trained on FISHBOWL dataset and adapt it to the KINS-Video-Car dataset at test-time. Under this setting,

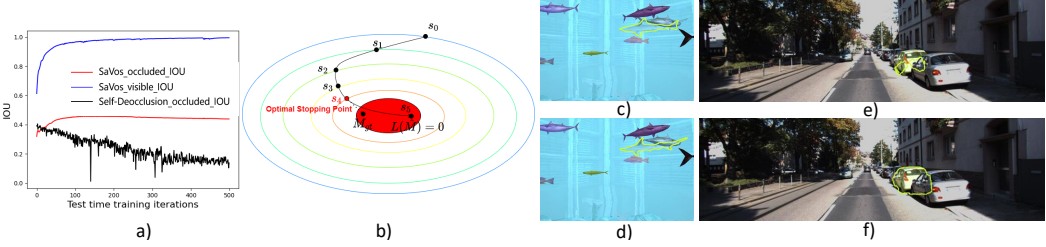

Figure 5: a) Learning curve for test-time adaptation on one FISHBOWL video. b) Learning dynamics for SaVos test-time adaptation. The optimal stopping point that is closest to the groundtruth might not be the one has the lowest loss. c) - d) SaVos first fits the unseen type of fish to a type seen in training time. After test-time adaptation, the incorrect type prior is gone. e) - f) Amodal performance before and after test-time adaptation for SaVos trained on FISHBOWL and test on KINS-Video-Car. In this case, though the model never see a car during training, it still make reasonable predictions. Though it captures less detail and slightly invades the neighboring pixels compared to a model trained on KINS-Video-Car in Figure 3 b).

Table 2: Methods with (w.) or without (w./o.) test-time adaptation (TTA). We report mean-IoU performance on unseen fishes. SaVos shows the advantage after applying test-time adaptation. S-D: Self-Deocclusion. Occ: occluded.

| Unseen | SmallFishA | MediaFishA | MediaFishB | BigFishA | BigFishB | Overall | |
|---|---|---|---|---|---|---|---|
| Full/Occ | Occ | Occ | Occ | Occ | Occ | Full | Occ |
| Self-D | 0.4757 | 0.6207 | 05825 | 0.5965 | 0.5709 | 0.8459 | 0.5859 |
| Self-D w. TTA | 0.4629 | 0.6349 | 0.6268 | 0.5742 | 0.5944 | 0.8497 | 0.5971 |
| SaVos w./o. TTA | 0.5362 | 0.6678 | 0.6044 | 0.5931 | 0.4978 | 0.8428 | 0.5905 |
| SaVos w. TTA | **0.5464** | **0.7147** | **0.7080** | **0.7151** | **0.6230** | **0.8663** | **0.6886** |
| SaVos w. All | 0.6299 | 0.7324 | 0.7256 | 0.7115 | 0.6894 | 0.8863 | 0.7155 |

Table 3: Adaptation performance from FISHBOWL training set to KINS-Video-Car test set. With test-time adaptation, SaVos trained on FISHBOWL achieves better mean-IoU compared with Self-Deocclusion trained on KINS-Video-Car.

| Method | Training | Adaptation | Full | Occluded |
|---|---|---|---|---|
| | FISHBOWL | ✗ | 0.7813 | 0.0658 |
| Self-Deocclusion | FISHBOWL | KINS-Video-Car | 0.7999 | 0.0969 |
| | KINS-Video-Car | ✗ | 0.8158 | 0.1790 |
| | FISHBOWL | ✗ | 0.8004 | 0.0994 |
| SaVos(Ours) | FISHBOWL | KINS-Video-Car | 0.8235 | 0.2976 |
| | KINS-Video-Car | ✗ | 0.8258 | 0.3132 |

the model is challenged to adapt to new object type, camera motion, new image quality as well as visual details. In Table 3, with test-time adaptation, our model even outperforms the image-level baseline trained directly on KINS-Video-Car, since our model leverages spatiotemporal prior, which leads to better generalization ability when distribution shift happens.

## 4.3 Analysis on The Effectiveness of Test-time Adaptation for SaVos

**Test-time adaptation optimization dynamic** Figure 5 a) shows the learning curve of test-time adaptation on one FISHBOWL video. The model is trained on other four types of fishes and tuned to adapt to the unseen type BigFishB in that video. For our SaVos model, we noticed the occluded IoU performance increases first and then slightly drops, while the mean-IoU on the visible part keeps improving. A similar observation has been made in Deep Image Prior [36] where on the image denoising task, the network first learns the denoised version of the input image and then reconstructs the noisy pixels later. We assume similar optimization dynamic on our case. As illustrated in Figure 5 b), the training loss in Equation 7 is a surrogate one that only optimizes the visible mask in the other frames. The ground-truth solution $M_{gt}$ belongs to a manifold of points that have loss: $L = 0$. However, there are other points can get the same loss while not really achieving perfect amodal segmentation. Usually those points refer to the predictions that not only cover the full mask, but also intrude into the neighboring area that is never revealed in any frame. Type prior can help resolving this issue by telling the common shape of that type. Then for seen types we don't worry about that. When type prior is not available in test-time adaptation, we try to tackle this using **early stopping**. We just stop when it finishes recovering all the visible part in every frame, leaving less chance to make additional intrusive predictions. In the experiment in this paper, we stop the optimization if the visible-part IoU improves less than 0.01 in the last 10 iterations. We also visualize the prediction before and after test-time adaptation in Figure 5 c)-f).

**Why test-time adaptation is efficient on SaVos?** Seems any self-supervised model is suitable in test-time adaptation since the same loss can be used in the training phase and test-time adaptation phase. However, we also try test-time adaptation on Self-Deocclusion while we don't observe the same amount of test performance improvement as SaVos in Table 2, 3.

The assumptions to explain this are *sampling efficiency* and *lack of motion*. Self-Deocclusion randomly selects another object as the occluder and overlay it on top of the visible mask of the target occludee to get supervision, again randomly. However, we would argue that such method is useful given a large training set but *particularly* not that efficient in test-time adaptation on a single video. The area they produces training signal is on the visible part at the current frame, which no need to be completed in this frame anyway. Only if the same part is visible in another frame and

the occluder happens to be overlapped on the same position, that the training signal is contributing to the current test time video. Considering the image resolution and occluder type, the chance to sample that training signal can be low. While for SaVos, we produce the training signal that is exactly on the occluded areas in this video by building amodal motion across frames. The efficiency on test-time adaptation setting is an unique perspective to compare different loss design. Though both are self-supervised, SaVos is more efficient than Self-Deocclusion after combining with test-time adaptation.

## 5 Conclusions

We propose SaVos, a self-supervised video amodal segmentation pipeline that simultaneously models the completed dense object motion and amodal mask. Beyond type prior, on which the existing image-level models rely, SaVos also leverages spatiotemporal priors for amodal segmentation. SaVos not only outperforms image-level baseline on several synthetic and real-world datasets, but also generalize better with test-time adaptation.

## 6 Limitations and Future Works

We summarize several limitations and future works of the existing SaVos model:

- Variational method can be introduced to handle uncertainty on the occluded area, especially for articulated non-rigid objects.
- Inductive bias from 3D modeling can be introduced to handle more complex ego and object motions.
- Currently, we need to run visible mask segmentation and tracking beforehand to start SaVos. It would be valuable to extend SaVos to an end-to-end pipeline, even all in self-supervised way. This leads to a future work of combining SaVos with video structured generative models like SCALOR[16]. However, we empirically tried to utilize the SCALOR code on KINS-Video-Car and found out object discovery on this dataset is still too challenging for existing methods. We'll also catch up with the progress of object discovery.

All these future works can be built on top of the idea of utilizing spatiotemporal information to mine amodal supervision signal and find evidence for mask completion from the existing SaVos.

## 7 Negative Social Impact

SaVos runs on object tracking result. Tracking on cars or pedestrians might have privacy concern.

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
