# Supplementary Material for Self-supervised Amodal Video Object Segmentation

**Jian Yao[1]\***    **Yuxin Hong[2]\***    **Chiyu Wang[3]\***    **Tianjun Xiao[4]†**    **Tong He[4]**
**Francesco Locatello[4]**    **David Wipf[4]**    **Yanwei Fu[2]†**    **Zheng Zhang[4]**

[1] School of Management, Fudan University
[2] School of Data Science, Fudan University
[3] University of California, Berkeley
[4] Amazon Web Services

{jianyao20, yxhong20, yanweifu}@fudan.edu.cn, wcy_james@berkeley.edu
{tianjux, htong, locatelf, daviwipf, zhaz}@amazon.com

We organize the Supplementary Material as follows:

- We conduct ablation studies on the designs of SaVos, showing how they come naturally with our insight to achieve amodal.
- As the necessity of $\mathcal{L}_C$ is observed, we have another dive into the consistency loss to explain its motivation.
- We provide theoretical analysis on the SaVos loss design, showing more insights on how SaVos get supervision signal by exploring spatiotemporal information.
- For the cases can not be handled by spatiotemporal prior, we use empirical experiments and analysis to verify that type prior can handle those.
- We provide details about the method for reproducing the experiments. Code is attached in the supplementary.
- We have more visualizations from SaVos model. Videos are also attached.

## A    Ablation Studies on The Designs of SaVos

### A.1    Ablation Study by Each Design Component

In this ablation study shown in Table 1, we verified the performance gain brought by each design component of SaVos on both FISHBOWL and KINS-Video-Car. The last row is the default setting for SaVos, according to the tables: removing any of the consistency loss, temporal embedding or bi-directional prediction will make the performance drop.

Note that our SaVos is not a combination of tricks each brings a little portion of gain. Those designs come naturally with our insight to achieve amodal completion using video. Since we use video, the consistency loss across frames is derived, and temporal embedding is also an intuitive design choice. Bi-directional prediction is not a must-have design. When we can run on offline setting, this is a why-not choice. Though SaVos already works well on online setting without Bi-directional prediction and the number beats the image-level baseline Self-Deocclusion[7]. We plot the mean value of the alpha channel for the forward pass and backward pass separately at each timestamp in Figure 1a). The plot meets our expectation of the alpha channel distribution: the alpha channel value for the forward pass is low at the beginning, increasing rapidly to get out of the "cold" region from frame 0-20, then slowing increasing from 20-100, finally increasing rapidly from 100-120 since that is the

---

\*Work completed during internship at AWS Shanghai AI Labs.
†Correspondence authors are Tianjun Xiao and Yanwei Fu.

Table 1: Ablation study on SaVos components

| Consistency Loss | Bi-directional prediction | Temporal Embedding | FISHBOWL | | KINS-Video-Car | |
| --- | --- | --- | --- | --- | --- | --- |
| | | | Full | Occluded | Full | Occluded |
| ✓ | ✓ | ✗ | 0.8785 | 0.6894 | 0.8214 | 0.2792 |
| ✗ | ✓ | ✓ | 0.8606 | 0.6447 | 0.8244 | 0.2982 |
| ✓ | ✗ | ✓ | 0.8749 | 0.6849 | 0.8231 | 0.2962 |
| ✓ | ✓ | ✓ | **0.8863** | **0.7155** | **0.8258** | **0.3132** |

Table 2: Ablations on the architecture and losses for SaVos

| Architecture | Loss | FISHBOWL | | KINS-Video-Car | |
| --- | --- | --- | --- | --- | --- |
| | | Full | Occluded | Full | Occluded |
| Self-Deocclusion [7] | Self-Deocclusion [7] | 0.8704 | 0.6502 | 0.8158 | 0.1790 |
| SaVos | Self-Deocclusion [7] | 0.8742 | 0.6826 | 0.8021 | 0.2040 |
| SaVos | SaVos | **0.8863** | **0.7155** | **0.8258** | **0.3132** |

"cold" region for backward pass. Combining predictions from both directions can help on the "cold start" issue.

## A.2 Disentanglement of The Contribution from Architecture and Self-supervised Loss

Compared with the ablation in Table 1 which focus more on the components of SaVos itself, this part compares with the related work by disentangling the influence of loss and architectures. According to Table 2, when both training with the self-supervision method in Self-Deocclusion [7], SaVos architecture introduces around 0.03-0.04 occluded mIoU performance gain. After replacing the loss into SaVos loss, the model achieves another 0.04 performance gain on FISHBOWL and 0.11 gain on KINS-Video-Car. Both architecture and loss contribute to the performance gain, while the loss contributes more for scenes like KINS-Video-Car.

## A.3 Ablation Study on SaVos Model Inputs

SaVos takes image patch $I$, visible mask $V$ and the optical flow patch for the visible part $\Delta V$ as inputs. As shown in Table 3, removing either flow input or image patch harms the performance. We analyze the results as follows:

**Removing optical flow input** Removing flow input hurts the performance by a large gap up to more than 0.1 occluded IoU. In SaVos model design, we need to warp the amodal prediction to the subsequent frame to get supervision. The warping function takes amodal flow as input. As described in supplementary Section E, we crop out the objects and rescale them to 64x128. This operation may lose the information of object motion. Then in the current setting, without flow, the network cannot infer object motion from the input thus the pipeline does not work properly. If we do not apply crop and scale, but put the full resolution mask and image as input, ideally the network can infer the flow from the sequence but directly providing the flow would avoid some unnecessary heavy-lifting and let the network focus on learning the completion signal.

**Removing image patch input** SaVos still outperforms the baseline in Self-Deocclusion[7]. Image patch provides important information about which part of the object is occluded now. The visible mask left the occluded part as background but image patch might contain the pixels of another object. Then the network would know where to complete. Image patch can provide some photomatic guidance for the flow completion, making the warping more accurate to collect more accurate supervision signals.

## A.4 Learning-based Versus Rule-based Spatiotemporal Modeling

SaVos learns spatiotemporal prior for amodal segmentation. A proper baseline is manually aggregating optical flow across frames into a complete object shape. The detailed algorithm is introduced in Algorithm 1. We also take the union of the forward and backward . As shown in the below Table 4, SaVos outperforms this temporal baseline.

Table 3: Ablations on the SaVos model inputs

| Model | FISHBOWL | | KINS-Video-Car | |
|---|---|---|---|---|
| | Full | Occluded | Full | Occluded |
| Without flow | 0.8508 | 0.6188 | 0.7922 | 0.1525 |
| Without image | 0.8673 | 0.6634 | 0.8025 | 0.2172 |
| Full | **0.8863** | **0.7155** | **0.8258** | **0.3132** |

Table 4: Learning-based SaVos versus rule-based spatiotemporal aggregation

| Model | FISHBOWL Occluded IoU | KINS-Video-Car Occluded IoU |
|---|---|---|
| Temporal aggregation baseline k=5 | 0.3860 | 0.1487 |
| Temporal aggregation baseline k=3 | 0.3937 | 0.1522 |
| SaVos | **0.7155** | **0.3132** |

The most important drawback to this naïve baseline is that if some parts of the objects are never visible in the whole video, this baseline of only using temporal information has no chance to complete the mask on those parts. This significantly limits the performance of this baseline in general. In contrast, SaVos proposes properly learning the type prior, which can still complete the whole mask.

To complete the whole object, this suggested naïve baseline may need to aggregate multiple frames. Consecutive warping across multiple frames might be inaccurate using the predicted flow. To verify this statement, we notice that k=3 actually works better than k=5, considering k=5 contains more information. In contrast, our SaVos model does not demand the constraint of consecutive warping, but only warping to one nearby frame. Our completion signals across multiple frames are aggregated by the consistency loss automatically using the backpropagation chain rule. This is the advantage of our learning based method that makes the model more robust to flow noise.

In the rule-based baseline, there won't be flow value for the occluded area for the occludee object. One has to employ a rule to fill the flow value for the occluded area; otherwise it is impossible to warp the whole to the next frame. To this end, this proposed baseline, we use the mean value of the visible part. However this might be inaccurate. In SaVos, we simultaneously predict the amodal mask and the flow value in the occluded area, which might be better than this baseline.

---

**Algorithm 1** Rule-based spatiotemporal aggregating method

---
**Input** $V, \Delta V$        $\triangleright$ Visible masks and flows for an object
**Initialize** $AV = V_{n-k}$        $\triangleright$ Aggregated visible mask as $AV$
**Initialize** $\Delta AV = \Delta V_{n-k}$        $\triangleright$ Aggregated optical flow as $\Delta AV$
**for** $t = n - k + 1$ to $n$ **do**
     $AV = Warp(AV, \Delta AV) \cup V_t$      $\triangleright$ Warp the aggregated mask to take the union with $V_t$
     $IV = (1 - V_t) \cap AV$      $\triangleright$ Identify the area completed by $AV$ but invisible in $t$
     $\Delta AV = \Delta V_t \cup (mean(\Delta V_t) * IV)$ $\triangleright$ Update $\Delta AV$, for the occluded part, use the mean to fill
**end for**

---

## B Another Dive into The Motivation of The Consistency Loss $\mathcal{L}_C$

One possible further enhancement on the supervision signal is to compare $\hat{M}_t^k$ to $V_l^k$ in all frames with $l \geq t$, considering different occluded areas might be revealed in different frames. In order to make such comparison, in principle we can supervise the model training with the following complete loss:

$$\mathcal{L}_{complete} = \sum_{k=1}^{K} \sum_{t=1}^{T-1} \sum_{l=t}^{T} d\left(\hat{M}_t^k, V_l^k\right) \quad (1)$$

With some distance measure $d$. However, in practice this loss has too high computational cost, and the distance measure between $\hat{M}_t^k$ and $V_l^k$ may be inaccurate when $l$ is far from $t$.

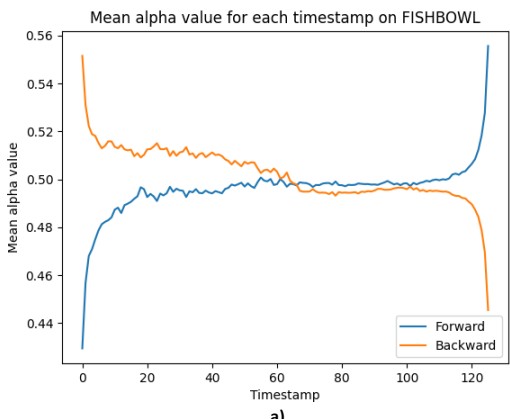
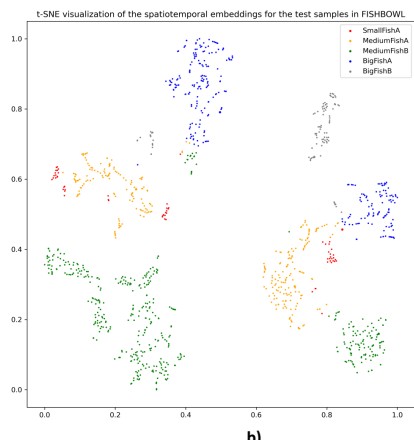

Figure 1: a) Distribution of the alpha channel of the bi-directional prediction. b) tSNE on the spatiotemporal embeddings for the test samples in FISHBOWL.

Instead of directly comparing $\hat{M}_t^k$ and all $V_l^k$, we build a chain to accumulate the training signals along the sequence using the chain rule. To be specific, based on the definition of $\mathcal{L}_M$ and $\hat{M}_{t+2}^k$:

$$d_M(\hat{M}_{t+2}^k, V_{t+2}^k) = d_M\left(\text{Warp}\left(\tilde{M}_{t+1}^k, \Delta\tilde{M}_{t+1}^k\right), V_{t+2}^k\right) \tag{2}$$

and

$$\hat{M}_{t+1}^k = \text{Warp}\left(\tilde{M}_t^k, \Delta\tilde{M}_t^k\right) \tag{3}$$

After adding the consistency loss $\mathcal{L}_C$, we have penalty on $d_C(\tilde{M}_{t+1}^k, \hat{M}_{t+1}^k)$. Then the supervision signal from $V_{t+2}^k$ will be passed through $\hat{M}_{t+2}^k \rightarrow \tilde{M}_{t+1}^k \rightarrow \hat{M}_{t+1}^k \rightarrow \tilde{M}_t^k$. So on so forth for $l > t+2$. This loss links the supervision signals in all future frames to guide the prediction in each frame.

## C   Theoretical Analysis on The Loss Design of SaVos

**Theorem 1** *Given $\mathcal{L}_M^k = \mathcal{L}_C^k = 0$, with the following assumptions, we claim that the region covered by $\hat{M}_t^k$ includes $M_t^k$: 1) visible masks and object motion are from ground truth, 2) object $k$ is rigid and simply connected, and 3) each pixel of this object is visible in at least one frame. Further, define an amodal trajectory $T_\mathbf{p}$ of point $\mathbf{p}$ as a sequence, $\{\mathbf{p}_t\}$ containing locations of a point $\mathbf{p}$ across frames, regardless of it being visible or occluded. We say $T_\mathbf{p} \parallel T_\mathbf{q}$ if $\mathbf{p}_t - \mathbf{q}_t$ is a constant for any $t$. Then, we claim that $\hat{M}_t^k = M_t^k$ with the following additional assumption: 4) Let $\mathbf{q}$ be an interior point of object $k$ with its trajectory $T_\mathbf{q}$, any exterior point, $p$, of object $k$ with its trajectory $T_\mathbf{p} \parallel T_\mathbf{q}$, appears in the background in at least one frame.*

This theorem makes an assumption that will not always strictly hold in practice, we emphasize that having global optima of the proposed loss is a necessary but not sufficient condition for aligning the correct amodal segmentation. The necessity indicates SaVos thoroughly captures the amodal supervision signals which could be explored from spatiotemporal information.

### C.1   Proof of Theorem 1

We start from notations, and formal definitions of rigid object and visibility. Next, we discuss the behavior under the two losses and their insufficiency. Finally, we prove the theorem.

**Notations** Given that $V_t^k$ represents the binary mask image for object $k$ at frame $t$, we further use $\mathcal{V}_t^k$ to refer to the set of pixels being one in $V_t^k$. Similarly, $\mathcal{M}_t^k$, $\hat{\mathcal{M}}_t^k$, and $\tilde{\mathcal{M}}_t^k$, are sets of pixels being one in $M_t^k$, $\hat{M}_t^k$, $\tilde{M}_t^k$ respectively.

Assuming a constant resolution for all $I_t$, let $\mathcal{U}$ be the set of all pixels in any image $I_t^k$. Since for any pair of objects $i$ and $j$ we have $\mathcal{V}_{t+1}^i \cap \mathcal{V}_{t+1}^j = \emptyset$, we define the background region $\mathcal{B}_t$ as

$$\mathcal{B}_t = \mathcal{U} \setminus \bigcup_{i=1}^{K} \mathcal{V}_t^i = \bigcap_{i=1}^{K} \overline{\mathcal{V}_t^i} \tag{4}$$

with $\overline{\mathcal{V}_{t+1}^i}$ as the complement of $\mathcal{V}_{t+1}^i$. With the above additional notations, for object $k$ with no occlusion at frame $t$ we have $\mathcal{M}_t^k = \mathcal{V}_t^k$. For an object $k$ being occluded at frame $t$ we have $\mathcal{V}_t^k \subset \mathcal{M}_t^k$, which means the visible region is a subset of the amodal region, and $\mathcal{M}_t^k \setminus \mathcal{V}_t^k \subseteq \bigcup_{i \neq k} \mathcal{V}_t^i$, which means the occluded region belongs to the visible region of other objects.

**Rigid object** Let object $k$ be a rigid object as in the assumption, its optical flow at any pixel is a pair of constant, specifically:

$$\begin{aligned}
\Delta V_{\mathrm{x},t}^k[x,y] &= \Delta v_{\mathrm{x},t}^k \\
\Delta V_{\mathrm{y},t}^k[x,y] &= \Delta v_{\mathrm{y},t}^k \\
\Delta M_{\mathrm{x},t}^k[x,y] &= \Delta m_{\mathrm{x},t}^k \\
\Delta M_{\mathrm{y},t}^k[x,y] &= \Delta m_{\mathrm{y},t}^k
\end{aligned} \tag{5}$$

for any $[x,y]$ in corresponding regions, where $\Delta v_{\mathrm{x},t}^k$, $\Delta v_{\mathrm{y},t}^k$, $\Delta m_{\mathrm{x},t}^k$, $\Delta m_{\mathrm{y},t}^k$ are four scalars for the optical flow, with $\Delta v_{\mathrm{x},t}^k = \Delta m_{\mathrm{x},t}^k$ and $\Delta v_{\mathrm{y},t}^k = \Delta m_{\mathrm{y},t}^k$.

As a result, its groundtruth amodal masks at frame $t$ and $t+1$ satisfy $M_{t+1}^k[x + \Delta m_{\mathrm{x},t}^k, y + \Delta m_{\mathrm{y},t}^k] = M_t^k[x,y]$. Generally, for any pixel $[x_t, y_t] \in \mathcal{M}_t^k$, we can warp it to its counterpart $[x_l, y_l]$ in frame $l$ since $[x_l, y_l] = [x + \sum_t^l \Delta m_{\mathrm{x},t}^k, y + \sum_t^l \Delta m_{\mathrm{y},t}^k]$.

**Visibility** The assumption on pixel visibility means that for any pixel $[x_t, y_t] \in \mathcal{M}_t^k$, there exists a frame $l$ and a pixel $[x_l, y_l] \in \mathcal{V}_l^k$ such that $V_l^k[x_l, y_l] = M_t^k[x_t, y_t]$ and $[x_l, y_l] = [x_t + \sum_t^l \Delta m_{\mathrm{x},t}^k, y_t + \sum_t^l \Delta m_{\mathrm{y},t}^k]$.

**Analysis on $\mathcal{L}_M$** We start with defining $\mathcal{W}_{t+1}^k$ as the set of pixels being one in $W_{t+1}^k$ from the equation: $W_{t+1}^k = \left(1 - \sum_{i=1}^K V_{t+1}^i\right) + V_{t+1}^k$.

By definition, we have

$$\mathcal{W}_{t+1}^k = \mathcal{B}_{t+1} \cup \mathcal{V}_{t+1}^k \tag{6}$$

Thus $\mathcal{L}_M$ can be re-written as

$$\mathcal{L}_M = \sum_{k=1}^{K} \sum_{t=1}^{T} \mathcal{L}_M^{(k,t)} \tag{7}$$

where

$$\begin{aligned}
\mathcal{L}_M^{(k,t)} &= \sum_{[x,y] \in \mathcal{W}_{t+1}^k} \mathrm{BCE}(\hat{M}_{t+1}^k[x,y], V_{t+1}^k[x,y]) \\
&= -\sum_{[x,y] \in \mathcal{B}_{t+1}} \log\left(1 - \hat{M}_{t+1}^k[x,y]\right) - \sum_{[x,y] \in \mathcal{V}_{t+1}^k} \log\left(\hat{M}_{t+1}^k[x,y]\right)
\end{aligned} \tag{8}$$

For any $k$ and $t$, $\mathcal{L}_M^{(k,t)}$ becomes zero only when:

$$
\hat{M}_{t+1}^k[x,y] = \begin{cases} 1 & \text{if } [x,y] \in \mathcal{V}_{t+1}^k \\ 0 & \text{if } [x,y] \in \mathcal{B}_{t+1} \\ arbitrary & \text{if } [x,y] \in \bigcup_{i \neq k} \mathcal{V}_{t+1}^i \end{cases} \tag{9}
$$

Note that for pixels in $\mathcal{V}_{t+1}^k$ and $\mathcal{B}_{t+1}$, the optimal value of $\hat{M}_{t+1}^k[x,y]$ equals to the true value in $M_{t+1}^k[x,y]$. However, for any pixel in $\bigcup_{i \neq k} \mathcal{V}_{t+1}^k$, $\hat{M}_{t+1}^k[x,y]$ can take arbitrary value without being penalized. Therefore $\hat{M}_{t+1}^k[x,y]$ is not guaranteed to exactly equal to $M_{t+1}^k[x,y]$, and this demonstrate the insufficiency of loss one.

**Analysis on $\mathcal{L}_C$** The assumption of rigid object simplifies the relation between $\tilde{M}_t^k$ and $\hat{M}_{t+1}^k$. Specifically the Warp operation becomes

$$
\hat{M}_{t+1}^k[x + \Delta \hat{m}_{\mathrm{x},t}^k, y + \Delta \hat{m}_{\mathrm{y},t}^k] = \tilde{M}_t^k[x,y] \tag{10}
$$

Again, we re-write $\mathcal{L}_C$ as

$$
\mathcal{L}_C = \sum_{k=1}^{K} \sum_{t=1}^{T-1} \mathcal{L}_C^{(k,t)} \tag{11}
$$

where

$$
\begin{aligned}
\mathcal{L}_C^{(k,t)} &= \sum_{[x,y] \in \tilde{\mathcal{M}}_{t+1}^k \cup \hat{\mathcal{M}}_{t+1}^k} d\left( \tilde{M}_{t+1}^k[x,y] - \hat{M}_{t+1}^k[x,y] \right) \\
&= \sum_{[x,y] \in \tilde{\mathcal{M}}_{t+1}^k \cup \hat{\mathcal{M}}_{t+1}^k} d\left( \hat{M}_{t+2}^k[x - \Delta \hat{m}_{\mathrm{x},t}^k, y - \Delta \hat{m}_{\mathrm{y},t}^k] - \hat{M}_{t+1}^k[x,y] \right)
\end{aligned} \tag{12}
$$

This formulation indicates that a zero $\mathcal{L}_C^{(k,t)} = 0$ means the masks in $\hat{M}_{t+1}^k$ and $\hat{M}_{t+2}^k$ have the same shape, but potentially at different locations. When $\mathcal{L}_C^{(k,t)} = 0$ for any frame $t \leq T$, we have that all $\hat{M}_t^k$ have the same shape. Formally it means for any $[x_t, y_t] \in \hat{\mathcal{M}}_t^k$ and any other frame $l$, there exists a pixel $[x_l, y_l] \in \hat{\mathcal{M}}_t^k$ such that $\hat{M}_l^k[x_l, y_l] = \hat{M}_t^k[x + \sum_t^l \Delta m_{\mathrm{x},t}^k, y + \sum_t^l \Delta m_{\mathrm{y},t}^k]$. We name this property as the Transitivity.

However, this loss doesn't constraint on the relation of $\hat{\mathcal{M}}_{t+1}^k$ to $\mathcal{V}_{t+1}^k$ or $\mathcal{B}_{t+1}$, thus it is not sufficient to recover $M_{t+1}^k$ by $\hat{M}_{t+1}^k$.

**Proof of $\mathcal{M}_t^k \subseteq \hat{\mathcal{M}}_t^k$** Assume $\left\{ \hat{M}_t^k \right\}_{t=1}^T$ is a set of amodal mask predictions for a rigid object $k$ that satisfies $\sum_t \mathcal{L}_M^{(k,t)} = 0$ and $\sum_t \mathcal{L}_C^{(k,t)} = 0$, we now proof $\mathcal{M}_t^k \subseteq \hat{\mathcal{M}}_t^k$ for any $t$.

Since $\sum_t \mathcal{L}_M^{(k,t)} = 0$, we have $\hat{M}_t^k[x,y] = M_t^k[x,y], \forall [x,y] \in \mathcal{V}_t^k \cup \mathcal{B}_t$. with $\sum_t \mathcal{L}_C^{(k,t)} = 0$, we also have the accurate optical flow scalars $\Delta \hat{m}_{\mathrm{x},t}^k = \Delta m_{\mathrm{x},t}^k$ and $\Delta \hat{m}_{\mathrm{y},t}^k = \Delta m_{\mathrm{y},t}^k$, by estimating the flow from pixels $[x_t, y_t] \in \mathcal{V}_t^k$ and $[x_{t+1}, y_{t+1}] \in \mathcal{V}_{t+1}^k$.

With the visibility assumption, for an occluded pixel $[x_t, y_t] \in \left( \bigcup_{i \neq k} \mathcal{V}_t^i \right) \cap \mathcal{M}_t^k$, we know there exists a frame $l$ such that $[x_t, y_t]$ can be warped to a visible pixel in another frame $l$. Combining this and the transitivity property when $\sum_t \mathcal{L}_C^{(k,t)} = 0$, we have $\hat{M}_t^k[x_t, y_t] = V_l^k[x_l, y_l] = \hat{M}_l^k[x_l, y_l]$. Since $[x_l, y_l] \in \mathcal{V}_l^k$, we know $\hat{M}_l^k[x_l, y_l] = M_l^k[x_l, y_l] = 1$. Thus, $\hat{M}_t^k[x_t, y_t] = M_t^k[x_t, y_t] = 1 \forall [x_t, y_t] \in \mathcal{M}_t^k$, and that means $\mathcal{M}_t^k \subseteq \hat{\mathcal{M}}_t^k$.

**Proof of $\mathcal{M}_t^k = \hat{\mathcal{M}}_t^k$** We have proved that $\mathcal{M}_t^k \subseteq \hat{\mathcal{M}}_t^k$. Here we further analyze the factors of having a non-empty $\hat{\mathcal{M}}_t^k \setminus \mathcal{M}_t^k$.

A pixel $[x_t, y_t] \in \hat{\mathcal{M}}_t^k \setminus \mathcal{M}_t^k$ cannot be in $\mathcal{B}_t$ since $\mathcal{L}_M = 0$, thus it has to be in the region $\left( \bigcup_{i \neq k} \mathcal{V}_t^i \right) \cap \overline{\mathcal{M}}_t^k$, i.e. on another object. With the transitivity property, there exists $[x_l, y_l] \in \hat{\mathcal{M}}_l^k$ such that $[x_l, y_l] = [x_t + \sum_t^l \Delta m_{x,t}^k, y_t + \sum_t^l \Delta m_{y,t}^k]$ in each frame $l$. Since $[x_t, y_t] \in \overline{\mathcal{M}}_t^k$, we have $[x_l, y_l] \notin \mathcal{M}_l^k$ for any frame $l$. As a consequence, for any frame $l$, we have similar conclusions as $[x_l, y_l] \in \hat{\mathcal{M}}_l^k \setminus \mathcal{M}_l^k$, $[x_l, y_l] \notin \mathcal{B}_l$, and $[x_l, y_l] \in \left( \bigcup_{i \neq k} \mathcal{V}_l^i \right) \cap \overline{\mathcal{M}}_l^k$.

On the other hand, for any pixel $[x_t, y_t]$, if it is in $\mathcal{B}_t$, or it could warp to a pixel $[x_l, y_l] \in \mathcal{B}_t$, then we have $[x_t, y_t] \notin \hat{\mathcal{M}}_t^k \setminus \mathcal{M}_t^k$, and also $[x_l, y_l] \notin \hat{\mathcal{M}}_l^k \setminus \mathcal{M}_l^k$ for all corresponding $[x_l, y_l] = [x_t + \sum_t^l \Delta m_{x,t}^k, y_t + \sum_t^l \Delta m_{y,t}^k]$. Therefore, if every pixel $[x_t, y_t] \in \overline{\hat{\mathcal{M}}_t^k}$ or any of its corresponding $[x_l, y_l]$ appears in the background, then $\hat{\mathcal{M}}_t^k \setminus \mathcal{M}_t^k = \emptyset$.

Because we made the assumption that no pixel out of $\mathcal{M}_t^k$ has relatively static motion to object $k$ across all frames that never appears in the background, so $\hat{\mathcal{M}}_t^k \setminus \mathcal{M}_t^k \neq \emptyset$ violates the assumption. Therefore, we conclude with $\mathcal{M}_t^k = \hat{\mathcal{M}}_t^k$.

## D    Empirical Analysis on How SaVos Handles Cases Missed by Spatiotemporal Prior

As mentioned in the above section, having global optima of the proposed loss is a necessary but not sufficient condition for aligning the correct amodal segmentation. To make good prediction on the cases where the assumptions in Theorem 1 are not hold, other prior and inductive bias from the architecture and data should chime in. To be more specific, we empirically found type prior learned through SaVos architecture handles cases missed by spatiotemporal prior. Typically, type prior models the shape prototype and variations for a certain type of object. When the object type is recognized, the model selects the prototype and certain variation to achieve amodal segmentation based on the context. The cases missed by spatiotemporal prior are usually the ones break the assumptions in Theorem 1. For example, part of an object keeps unseen in the whole video, as shown in Figure 2, where the car marked in blue behind the closest-in-path-vehicle (relative to ego) has the bottom left part always invisible. SaVos still complete the full amodal mask of the whole car.

Quantitatively, we split KINS-Video-Car into two parts. One presumably contains cases that break the "pixel-wise visible in video" assumption. Specifically, we collect 2899 out of 4644 cases of which the visible masks are touching with other objects in all tracked frames. This criterion roughly picks out the desired cases (roughly 70% of the selected cases break the assumption). In Table 5, the mIoU on the occluded part for this subset is not significantly lower than the numbers for the whole set and the other subset. This validates SaVos has the capacity of handling more general amodal scenarios.

From the perspective of the architecture inductive bias, the encoder-decoder architecture in the Amodal Maks Completor contains an information bottleneck. This makes it easy to squeeze out type information since type is a concise representation and beneficial to amodal prediction. By receiving amodal supervision signals on different parts from different scenes, as long as the model learns to know they are amodal signals for the same type, those signals will be accumulated to form the type prior. This design choice also appears in several image-level methods such as [7] and [3]. In that case, SaVos can work as well as image-level methods on the cases break the "pixel-wise visible in video" assumption as long as the part invisible in this video can show up on different instance for the same type in other videos. If such a loosened assumption does not hold, then there will be no supervision signal for that part in the whole dataset, image-level baseline models will also fail. Empirically, this can be verified in Table 1. SaVos can beat the Self-Deocclusion[7] baseline even without Temporal Embedding module. In test-time adaptation scenario where new types of objects appear, the methods rely only on type prior will fail while SaVos still have the chance to provide amodal completion using spatiotemporal prior.

To further verify this statement, We run tSNE on the spatiotemporal embeddings for the test samples FISHBOWL in Figure 1b) and noticed the data shows a clear clustering pattern consistent with the type information. Though objects for the same type might be split into more than one clusters, data points from different classes usually won't be entangled. This is an evidence that type information are learned.

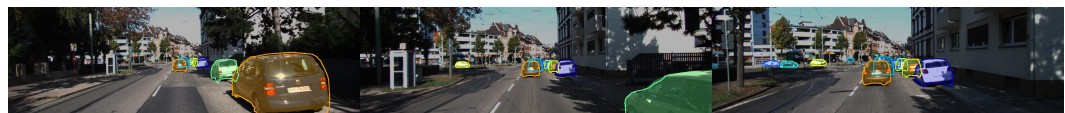

Figure 2: The car marked in blue behind the closest-in-path-vehicle (relative to ego) has the bottom left part always invisible. SaVos still complete the full amodal mask of the whole car. This is an indicator that SaVos not only just learn spatiotemporal prior, but also type prior.

Table 5: Occluded-mIoU on subsets of KINS-Video-Car

|  | Full set | Subset 1 (break the assumption) | Subset 2 |
|---|---|---|---|
| mIoU on occluded part | 0.3132 | 0.3104 | 0.3250 |

# E   Additional Details about the Method

## E.1   Detailed Architecture Hyperparameters

We use CNN to encode visual input for Spatiotemporal Embedding Module, Amodal Completer and Motion Predictor. The CNN has 6 layers. Each layer is followed by Group Normalization[5] and leaky ReLU nonlinearity. The default hyperparmeters for CNN encoder are given in the following Table 6.

To aggregate features on the temporal dimension, we use LSTM with 256 hidden dimensions in the Spatiotemporal Embedding Module.

Table 6: Default hyperparameters for CNN encoders

| Parameter | Setting |
|---|---|
| internal resolution | $64 \times 128$ |
| channels in hidden layers | 32, 64, 128, 256, 256, 512 |
| strides in hidden layers | 2, 2, 2, 2, (1,2), 1 |
| kernels in hidden layers | 4, 4, 4, 4, (3,4), 4 |
| padding in hidden layers | 1, 1, 1, 1, 1, 0 |

The decoders in Amodal Completor and Correspondence Predictor are built symmetrically by using the reversed list of transposed convolutions and layer parameters. Additionally for the last layers we remove the normalization and Leaky ReLU and add the sigmoid nonlinearity.

## E.2   Implementation Details

We implement our model using PyTorch [4], which has BSD-style license. We use FlowNet2 [1] to extract optical flow. For all datasets, We train the model with the Adam [2] optimizer with the learning rate of 0.0001. We train 20 epochs on Chewing Gum and FISHBOWL dataset, 200 epochs for KINS-Video-Car dataset and remain the model with the lowest self-supervised loss on the validation set. The batch size are 64, 24 and 8 respectively. We train on an AWS g4dn.12xlarge EC2 instance with four T4 GPUs. The training time is about one hour for Chewing Gum dataset and 24 hours for both FISHBOWL and KINS-Video-Car dataset.

## E.3   mIoU Metric Computing

For each frame in a video, we calculate the mean IoU of the objects in that frame, and then average over all frames in a video dataset. We use this frame-level mean IoU metric in the paper. We can also calculate the mean IoU for all the objects without considering frames, which can be referred to as object-level mean IoU. We provide the results obtained by the above two metric computing methods in Table 7, 8 and comparing with the baseline, showing that the difference of these two metrics is

Table 7: Frame-level mean IoU

| Method | FISHBOWL | | KINS-Video-Car | |
|---|---|---|---|---|
| | Full | Occluded | Full | Occluded |
| Self-Deocclusion [7] | 0.8704 | 0.6502 | 0.8158 | 0.1790 |
| SaVos (Ours) | **0.8863** | **0.7155** | **0.8258** | **0.3132** |

Table 8: Object-level mean IoU

| Method | FISHBOWL | | KINS-Video-Car | |
|---|---|---|---|---|
| | Full | Occluded | Full | Occluded |
| Self-Deocclusion | 0.8678 | 0.6418 | 0.7931 | 0.1952 |
| SaVos (Ours) | **0.8843** | **0.7111** | **0.8063** | **0.3312** |

small and the conclusions made in this paper are hold in either metric. Our released code provides the computing for both metrics.

## F  More Visualizations from SaVos

### F.1  Visualization on KINS-Video-Car

The visualization on KINS-Video-Car in Figure 3 is mainly to show SaVos's performance on cars with different viewing angles and occlusion patterns. KINS-Video-Car is particularly challenging considering there is ego-camera model. Also, even use off-the-shelf SOTA segmentation and tracking model, the predicted visible mask and tracking still can't be perfect. SaVos still be able to complete object mask from different views of the cars.

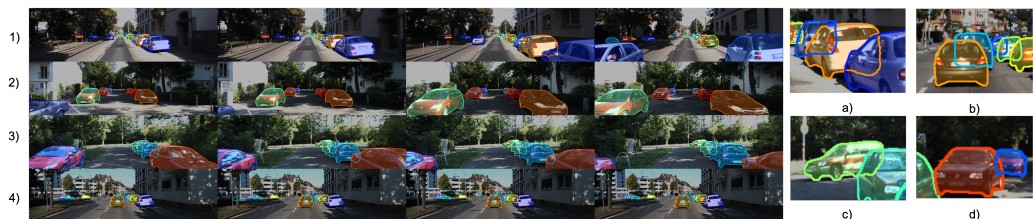

Figure 3: Visualization of the prediction of our SaVos model on KINS-Video-Car dataset. The transparent mask is the predicted modal mask by PointTrack[6]. The solid curve are the contours of the amodal prediction by SaVos. From the figure we can see our model can predict the amodal mask well in the scene with multiple cars and heavy occlusion. Our model performs well for cars in different viewing angles and occlusion patterns. For examples: a)-d), which are cropped from the case 1)-4), respectively.

### F.2  Visualization on Chewing Gum

Chewing Gum is a classless dataset. The image-level models fail since there is no type prior to be learned, even for the supervised method. SaVos learns to aggregate the temporal information from the video to achieve amodal mask completion in this dataset.