# OpenReview forum: "Self-supervised Amodal Video Object Segmentation"
_NeurIPS.cc/2022/Conference — NeurIPS 2022 Accept_

### Official Review · Reviewer_w3Qn · 2022-07-08

**Rating:** 6
**Confidence:** 5
**Soundness:** 3 good
**Presentation:** 3 good
**Contribution:** 3 good

**Summary:**

The paper proposes a self-supervised method (SaVos) for amodal segmentation that tracks shapes of partially occluded objects in video frames to learn the complete (amodal) shape. Given visible modal segmentation masks, the method computes amodal masks and objects’ motion by relying on both spatiotemporal and type priors. The method generalizes well by applying one-shot test-time adaptation given that the focus is on the spatiotemporal prior in order to retrieve the output. The type prior is however useful in scenarios where some parts of the objects are never visible.

**Questions:**

-

**Limitations:**

The paper does not address the limitaitons of the method.

**Strengths And Weaknesses:**

Strengths

+ The paper is very well written paper. The explanations are complete and make the paper is easy to follow.
+ Amodal segmentation is an arduous task and the key ideas to solve it are very intuitive and relevant.
+ Although the method is a combination of well known techniques, it is valuable in the the sense that there are improvements compared to the presented SotA even after a drastic distribution shift.
+ The connection to human vision in the first paragraph of the introduction is inspiring and informative.

Weaknesses

- The related work is not adequately mentioned or compared with. It should at least include [A] Ling et al. (NeurIPS 2020) and [B] Sun et al. (CVPR 2022). In the experiments, the baseline used to compare the results should be Ling et al. in addition to Zhan et al. (CVPR 2020), since it is a newer and better performing model. Moreover, similarly to what has been done in Sun et al., it would be more informative to present results as a function of the different occlusion levels, which would enable studying the effect of increasing amount of partial occlusion.
- In the experiments, the authors do not compare to any method that uses temporal information, which gives the proposed model an important advantage over related work. The authors state in lines 41-42 “traditional optical flow […] require[s] explicit visual correspondence across pixels, which would be impossible due to occlusions”. However, I actually think that aggregating optical flow across frames into a complete object shape would be the proper baseline for the proposed mode (also including the input provided, i.e. assuming rigid objects, slowly moving object, ground-truth bounding box and visible segmentation)?
- The proposed method makes very strong assumptions regarding the supervision at training time. In particular, the model assumes that the object is perfectly discovered, including ground-truth bounding box, instance label and visible segmentation mask. Therefore, referencing this approach as self-supervised is somehow misleading. Moreover, these assumptions are rather unrealistic in real world scenarios, therefore the experiments should study how sensitive the proposed method when those inputs are predicted, e.g. using a Faster-RCNN model.
- The authors state in lines 275-276 “occluded part only occupies a small portion of the entire full mask, thus mean-IoU on the occluded part is a better indicator on the model performance”: This does not mention that this indicator is also significantly more sensitive to noise. Results should include standard deviation estimates to ensure that the uncertainties are considered when comparing results.
- It would be worthwhile to see how does this method generalizes to additional categories i.e. see how much the one-shot test-time adaptation to a new domain “unlearn” previous learned type priors, e.g. performing training on FISHBOWL, adaptation on KINS-Video-Car and re-evaluation on FISHBOWL would give enlightening clues about the generalization capacity of the method.
- The authors state in the Supplementary material lines 140-141: “this makes it easy to squeeze out type information since type is a concise representation”: Were there any experiments performed to support this statement? It would be interesting to underline the possibility to apply classification or clustering to see if it is possible to extract type information as it was mentioned.
- On what grounds are these statements based on? “reasonably learned” (l. 8-9), “as long as the deformation can be reasonably learned” (l. 56), “This module also learns reasonable deformation over time” (l. 216)
- The paper mentions that the SAIL-VOS dataset is not used because it “has frequent camera switches” (l.79). It would still be interesting to study the performance of the model on that dataset. How long does a video sequence need to be for the proposed model to perform well compared to related work?

Minor:

- Concerning the alpha channel in the bi-direction prediction, does it help only at the beginning when the cold start problem is present or is it also useful at any point after? An analysis of the weights distribution of this channel to further detail its influence would be a great added value.
- Reference to figures, equations and sections (especially in sec. 3.1) are inconsistent and sometimes missing (l. 175 and l.178). Please make sure to precise what you are referring to in order to ensure clarity.
- The paper mentions that a limitation of related works is that they need to perform object discovery (l.  87.). At that location in the text it should be mentioned that this work assumes that the object is perfectly discovered, including ground-truth bounding box, instance label and visible segmentation mask.

============================ POST REBUTTAL==========================
After reading the other reviews and the rebuttal I increase my score to 6 and vote for accepting the paper. I very much appreciate the detailed comments of the authors which addressed my concerns and provide important clarifications regarding the points that I misunderstood. I think this is a solid work with moderate contribution. Remaining weaknesses to me are that it was mostly studied on rather simple video datasets and does not generalize to image collections such as COCO-A or OccludedPASCAL3D+ that are standard evaluation benchmarks in related works, due to the necessity for video data.

---

> ### Author Response · Authors · 2022-08-02
> **Response to Reviewer w3Qn Part 1**
>
> We thank the reviewer for the detailed review as well as the suggestions for improvement. We are also preparing the new version of the paper based on the suggestions and additional experiments. Our responses to the reviewer’s comments are below.
>
> **1.Comparison with [A] Ling et al. (NeurIPS 2020) and [B] Sun et al. (CVPR 2022).**
>
> Thanks for suggesting those related works. We will cite them. We run the code for Sun et al. (CVPR 2022) and here is the comparison with SaVos on KINS-Video-Car dataset on the full mIoU/occluded mIoU at different occlusion ratios:
>
> |Occlusion ratio | 0.7-1.0 | 0.3-0.7 | 0-0.3 | Overall |
> |------------------------|---------------|---------------|---------------|---------------|
> | Sun et al. (CVPR 2022) | **0.5744**/0.2765 | 0.5886/0.1738 | 0.8506/0.1877 | 0.7837/0.1897 |
> | SaVos | 0.4968/**0.3919** | **0.6579/0.4520** | **0.8657/0.2644** | **0.8270/0.3159** |
>
> We could notice SaVos performs better compared with Sun et al. (CVPR 2022).
>
> We didn’t conduct the comparison with Ling et al. (NeurIPS 2020) for the following reasons:
> 1.  Ling et al. (NeurIPS 2020) uses the same strategy as Zhan et al. (CVPR 2020) to generate amodal supervision.
> 2.  The code for Ling et al. (NeurIPS 2020) is not open-sourced. We resplit the KINS dataset to run video amodal task (test videos in the original split are not available). Thus the reported KINS results in Ling et al. (NeurIPS 2020) are not comparable. Without code open-sourced, we also can’t rerun the model.
>
> **2.Aggregating optical flow across frames into a complete object shape as a proper baseline uses temporal information.**
>
> Thanks for this great suggestion! Indeed, there is no existing video amodal work that can serve as a baseline for explicitly exploiting temporal information. The suggested baseline is a naive but intuitive baseline to better justify our method, while it does have many drawbacks, as discussed below. We set up the baseline as follows:
>
> For an object in a certain frame $n$, we aggregate the visible masks and optical flow from frame $n- k$ to $n$.
>
> Initialization:
>
> Aggregated visible mask as $AV$, initialized as $AV=V_{n-k}$
>
> Aggregated optical flow as $\Delta AV$, initialized as $\Delta AV=\Delta V_{n-k}$
>
> for $t$ in range($n-k+1$, $n+1$):
>
> &nbsp;&nbsp;&nbsp; $AV = Warp(AV, \Delta AV) \cup V_{t}$ # Warp the aggregated mask into frame t and take the union with $V_{t}$ to update $AV$
>
> &nbsp;&nbsp;&nbsp; $IV = (1 - V_{t}) \cap AV$ # Identify the area completed by $AV$ but invisible in $t$
>
> &nbsp;&nbsp;&nbsp; $\Delta AV = \Delta V_{t} \cup (mean(\Delta V_{t}) * IV)$ # Update $\Delta AV$, for the occluded part, use the mean value of the visible flow to fill.
>
> We also take the union of the forward and backward $AV$. As shown in the below table, SaVos outperforms this temporal baseline.
>
> | Model                                         | FISHBOWL Occluded IoU | KINS-Video-Car Occluded IoU |
> |-----------------------------------------------|-----------------------|-----------------------------|
> | Temporal aggregation baseline k=5 | 0.3860                | 0.1487                      |
> | Temporal aggregation baseline k=3 | 0.3937                | 0.1522                      |
> | SaVos                                         | **0.7111**            | **0.3159**                  |
>
> We give some discussions and the justifications of this experiment:
>
> 1.  The most important drawback to this naïve baseline is that if some parts of the objects are never visible in the whole video, this baseline of only using temporal information has no chance to complete the mask on those parts. This significantly limits the performance of this baseline in general. In contrast, our SaVos proposes properly learning the type prior, which can still complete the whole mask.
>
> 2.  To complete the whole object, this suggested naïve baseline may need to aggregate multiple frames. Consecutive warping across multiple frames might be inaccurate using the predicted flow. To verify this statement, we notice that k=3 actually works better than k=5, considering k=5 contains more information. In contrast, our SaVos model does not demand the constraint of consecutive warping, but only warping to one nearby frame. Our completion signals across multiple frames are aggregated by the consistency loss automatically using the backpropogation chain rule as described in L160-163. This is the advantage of our learning based method that makes the model more robust to flow noise.
>
> 3.  In the suggested baseline, there won’t be flow value for the occluded area for the occludee object. One has to employ a rule to fill the flow value for the occluded area; otherwise it is impossible to warp the whole $AV$ to the next frame. To this end, this proposed baseline, we use the mean value of the visible part. However this might be inaccurate. In SaVos, we simultaneously predict the amodal mask and the flow value in the occluded area, which might be better than this baseline.

---

> > ### Author Response · Authors · 2022-08-02
> > **Response to Reviewer w3Qn Part 2**
> >
> > **3.The model assumes GT input. The experiments should study how sensitive the proposed method when those inputs are predicted. Also, referencing this approach as self-supervised is somehow misleading.**
> >
> > Thanks. We clarify the misunderstanding from the reviewer that the inputs (including all the instance labels (track ID), optical flow and visible segmentation mask) for the KINS-Video-Car are predicted from the network, rather than using the ground-truth. Specifically, as described in L240, “We use PointTrack [40] to extract visible masks and object tracks to drive our video-based alg”. PointTrack ([https://github.com/detectRecog/PointTrack](https://github.com/detectRecog/PointTrack)) is a SOTA method for multiple object tracking and segmentation. And for optical flow, we use FlowNet2 ([https://github.com/NVIDIA/flownet2-pytorch](https://github.com/NVIDIA/flownet2-pytorch)). In that sense, we clarify that our model is self-supervised on the amodal mask completion, as we do not employ any labeled occluded parts from human annotators. This follows Zhan et al. (CVPR 2020), they pretrained an UNet to get modal segmentation supervisedly and the title for their paper is “Self-Supervised Scene De-occlusion”. We will emphasize self-supervision is only for the amodal mask completion in the later version.
> >
> > **4.mean-IoU on the occluded part is a better indicator but might also be significantly more sensitive to noise.**
> >
> > Thanks for this point. We explained in L253 that “The performance difference between runs is around 0.02 on the occluded mean-IoU.” We highlight that on all datasets, all the compared methods on the occluded mean-IoU, have round similar standard deviations, which are significantly smaller than the performance difference of these methods.
> >
> > **5.How does this method generalizes to additional categories i.e. see how much the one-shot test-time adaptation to a new domain “unlearn” previous learned type priors.**
> >
> > Thanks for the comments. In the one-shot test-time adaptation phase, the old type prior prediction and spatiotemporal prior prediction will conflict on unseen categories. The SaVos loss encourage spatiotemporal prior learning, then in this phase, the model will "unlearn" the old type prior and only use spatiotemporal prior to complete. The proposed experiment is indeed an interesting deep dive on this. We run the experiment in the following setting: we run one-shot test-time adaptation on one KINS-Video-Car test sample and save the model of different test-time iterations, then evaluate those models on the whole FISHBOWL test set. As the below table shows, the “unlearn” happens quickly at the first few iterations, then the performance gradually becomes stable.
> >
> > Train on FISHBOWL, test-time adapted on 1 KINS-Video-Car video and retest on FISHBOWL
> >
> > | Test Time Adaption Iterations| 0 | 1 | 6 | 11 | 16 | 21 | 26 |
> > |:------------------------:|:------:|:------:|:------:|:------:|:------:|:------:|:------:|
> > | Full IoU | 0.8874 | 0.8853 | 0.8667 | 0.8600 | 0.8574 | 0.8564 | 0.8557 |
> > | Occluded IoU | 0.7111 | 0.7097 | 0.6495 | 0.6128 | 0.6031 | 0.6022 | 0.5990 |
> >
> > **6.Experiments performed to support type information are learned.**
> >
> > Thanks for the suggestion. We added a visualization experiment during the author response phase to verify this as suggested. To be more specific, We run tSNE on the spatiotemporal embeddings for the test samples FISHBOWL and noticed the data shows a clear clustering pattern consistent with the type information. Though objects for the same type might be split into more than one clusters, data points from different classes usually won't be entangled. This is an evidence that type information are learned. Please check the visualization through this anonymous link: [https://ibb.co/wQGpfnj](https://ibb.co/wQGpfnj)
> >
> > **7. Verify “This module also learns reasonable deformation over time”**
> >
> > Thanks for the suggestion. The deformation caused by viewing angle change is common in the real-world KINS-Video-Car dataset. While the ego vehicle is moving forward, the mask shapes of the same car at the different frames are different. As shown in this anonymous link: https://ibb.co/vw6gB3d , SaVos can complete the amodal mask differently from different viewing angles. This is an indicator that the model learns deformation.

---

> > > ### Author Response · Authors · 2022-08-02
> > > **Response to Reviewer w3Qn Part 3**
> > >
> > > **8. Experiments on SAIL-VOS**
> > >
> > > Thanks for the suggestion. In general, we had conducted extensive experiments on several datasets to show the efficacy of our method, while the experimental results are extensive enough to support our contributions. We do notice the SAIL-VOS dataset which is a great “movie style” dataset for video based tasks. Unfortunately, we also found that this dataset has frequent camera switches, which makes it not the ideal testbed for our task. We give more explanations here: the simulated data in SAIL-VOS dataset is the consumer videos, or the “movie style” videos where the camera is focusing on the characters for story narratives. So this will cause several problems (can be shown in the examples in the anonymous link [https://ibb.co/P4M8PxT](https://ibb.co/P4M8PxT) ): 1. Consecutive frames might be taken in totally different views as the example in the link. This will break object tracking and optical flow. This happens frequently, almost every 1-2 seconds; 2. Since the camera is focusing on the characters, the objects might be in arbitrary scale and pose. For example, for the car in frame000018 in the link, major part of the car is out of the field of view of the camera. This will break most of the amodal segmentation network where the model assumes the object is in the field of view. Thus, since our model takes the advantage of the inductive bias from the spatiotemporal modeling and the common-fate principle of Gestalt Psychology (L60-63), this dataset may not be the good testbed for our model.
> > >
> > > On the other hands, the dataset settings in our paper might be more suitable for some amodal segmentation use scenarios like autonomous driving and embodied robotic.
> > >
> > > **9. limitations of the method are not addressed.**
> > >
> > > We discussed the limitation and social impact in the supplementary Section G and H. We will move that part into the main doc in the later version. Thanks for pointing this out.
> > >
> > > **Minor Points:**
> > >
> > > **M1 .Distribution of the alpha channel of the bi-directional prediction.**
> > >
> > > Thanks for the suggestion. We plot the mean value of the alpha channel for the forward pass and backward pass separately at each timestamp as suggested in the anonymous link: [https://ibb.co/ZdwW5Gs](https://ibb.co/ZdwW5Gs) . The plot meets our expectation of the alpha channel distribution: the alpha channel value for the forward pass is low at the beginning, increasing rapidly to get out of the “cold” region from frame 0-20, then slowing increasing from 20-100, finally increasing rapidly from 100-120 since that is the “cold” region for backward pass.
> > >
> > > **M2.The paper mentions that a limitation of related works is that they need to perform object discovery (l. 87.). At that location in the text it should be mentioned that this work assumes that the object is perfectly discovered.**
> > >
> > > Thanks. We clarify the misunderstanding of the reviewer on this point. As discussed in the third point, all the instance label (track ID), optical flow and visible segmentation mask for the KINS-Video-Car are predicted from network, not groundtruth. Self-supervised object discovery is a major challenge on those structured generative model or object-centric model. Those models are not focusing on making amodal mask completion work at real world but more on scene decomposition into object-centric representation. We’ll add more discussions in the revision. We highlight that these works are orthogonal and potentially useful to SaVos. When object discovery and object-centric learning makes progress on real-world videos, we can consider combining SaVos with object-centric learning as future works.
> > >
> > > **M3.Typos**
> > >
> > > Thanks very much for catching these. We will fix them in the later version.

---

> > > > ### Public Comment · ~Zhixuan_Li1 · 2022-11-29
> > > > **Thanks for your explanation of not using the SAIL-VOS dataset**
> > > >
> > > > Thanks for your detailed explanation. I have been developing a new method for the video-level AIS problem on the SAIL-VOS dataset for several months. My proposed video-level method still cannot beat the image-level method on the SAIL-VOS dataset, and I didn't find the reason. Your explanation and illustration of the viewpoint switch tackle my question.
> > > >
> > > > Do you have a plan to make the KINS-Video-Car dataset publicly available?
> > > >
> > > > And could you give a link to the Chewing Gum Dataset?
> > > >
> > > > Thank you so much!

---

> > > > > ### Public Comment · Authors · 2022-11-30
> > > > > **The code preparation is ongoing.**
> > > > >
> > > > > The code will be released soon, including the KINS-Video-Car dataset and the script to generate the Chewing Gum. Feel free to contact us if you have any questions!

---

> ### Author Response · Authors · 2022-08-08
> **Looking forward to your further feedback (Reviewer w3Qn)**
>
> Thanks again for the detailed review as well as the suggestions for improvement.
>
> We tried our best to address each of the comments and suggestions: we run additional comparison and ablation experiments (**1,2,5**), clarify and emphasize certain parts in the paper that might cause misunderstanding (**3,4,9,M2**), add visual analysis to support statements in the paper (**6,7,M1**), and add further justification on our experiment dataset selection (**8**).
>
> We sincerely look forward to your further feedback to see if the response addresses your concerns. Thanks!

---

### Official Review · Reviewer_CKtM · 2022-07-10

**Rating:** 7
**Confidence:** 4
**Soundness:** 3 good
**Presentation:** 3 good
**Contribution:** 3 good

**Summary:**

The paper proposes a method for amodal video object segmentation that does not rely on amodal segmentation annotations during training. Starting with image, optical flow and conventional segmentation as input, the method estimates an amodal mask and amodal motion for each instance and propagates this mask through time using the estimated flow. The learning signal comes from consistency terms with the input mask and flow through time.

**Questions:**

Questions
The questions follow the weaknesses above.
1. How does the paper position itself compared to the related work above? This includes a potential discussion on unsupervised vs. weakly supervised methods.

2. What is the influence of the accuracy of the input optical flow and segmentation on the  performance of the method?

3. It is somewhat difficult to attribute the performance gains over [42] as the method uses a different architecture as well as a different formulation. What influence do the proposed architecture changes have on the final performance?

4. In table 1, why is the supervised baseline called “supervised oracle”? It is useful to compare to a supervised baseline, but what makes it an oracle in addition?


**Limitations:**

Some limitations are discussed in the appendix. It could be helpful to expand them with actual examples from the results.
Societal impact is discussed only very briefly: one line in the appendix. This could be expanded considerably. It is important to discuss that these models often hallucinate segmentations of parts of objects that are never actually visible and thus cannot be trusted.

**Strengths And Weaknesses:**

Strengths
The paper has a clear motivation for learning amodal segmentation without amodal supervision. The writing is clear and the explanations are comprehensive.

The method itself is intuitive, by using the assumption that cumulatively, over the course of a video most of the object will be visible.

The results show good performance as well as the importance of test-time fine-tuning and generalization to unseen objects and sequences.


Weaknesses
Positioning and comparisons
The paper is missing a discussion and positioning relative to many recent works in the area:
-Amodal segmentation
Nguyen, Khoi, and Sinisa Todorovic. "A weakly supervised amodal segmenter with boundary uncertainty estimation." ICCV 2021, performs image-based amodal segmentation.
Ling, Huan, et al. "Variational amodal object completion." NeurIPS 2020, does amodal completion per object instance in single frames.
Zhou, Qiang, et al. "Human De-occlusion: Invisible Perception and Recovery for Humans." CVPR 2021, learns occlusion for people.
Zheng, Chuanxia, et al. "Visiting the Invisible: Layer-by-Layer Completed Scene Decomposition.", IJCV 2021
-Weakly supervised instance segmentation
Arun, Aditya, C. V. Jawahar, and M. Pawan Kumar. "Weakly supervised instance segmentation by learning annotation consistent instances." ECCV 2020
The setup is quite similar to weakly supervised amodal segmentation - of course here applied to videos instead of images - so it would make sense to also classify this work as weakly supervised (uses segmentation and flow input) instead of unsupervised.

Influence of input accuracy
As the method uses optical flow and instance segmentation input, it would be interesting to understand the dependency of the method on the accuracy of that input. Instead of using the GT on synthetic datasets, one could utilize a sota (supervised) method to estimate flow and instance segmentation and compare the results to GT inputs. As the paper is making a case for _unsupervised_ amodal segmentation it could be achieved through the use of unsupervised instance segmentation and unsupervised optical flow methods, however likely at the cost of performance as these methods fall behind their supervised counterparts.

---

> ### Author Response · Authors · 2022-08-02
> **Response to Reviewer CKtM Part 1**
>
> We thank the reviewer for the detailed review as well as the suggestions for improvement. We are also preparing the new version of the paper based on the suggestions and additional experiments. Our responses to the reviewer’s comments are below.
>
> **1.Related works and discussion on the terms use of “self-supervised” or “weakly supervised”**
>
> Thanks for providing those related works. We will cite them in a later version. Nguyen et.al (ICCV 2021) and Zhan et al. (CVPR 2020) get the supervision signal in a similar way with Zhan et al. “Self-supervised scene de-occlusion” (CVPR 2020). Their innovation parts are on the different inductive biases in the architecture design. Zhou, Qiang, et al (CVPR2021) learns amodal completion for humans using a supervised method by collecting a labeled dataset. Zheng, Chuanxia, et al. ( IJCV 2021) presents a layer-by-layer network which iteratively performs instance segmentation and scene competition using simulation. Aditya Arun(ECCV2020) proposes a weakly supervised method on image instance segmentation, utilizing class labels. Critically, our model exploits self-supervised learning to model spatio-temporal cues of videos, which is different from the suggested papers.
>
> For experiment comparison, we select Zhan et al. “Self-supervised scene de-occlusion” (CVPR 2020) as the major baseline to compare self-supervision formulation. As mentioned above, some of the works above are using the same or similar method with Zhan et al. (CVPR 2020) to get supervision, e.g.  Ling, Huan, et al. (NeurIPS 2020). Architecture-wise, we find and compare with a recent paper claiming SOTA, “Amodal segmentation based on visible region segmentation and shape prior” (AAAI 2021), and it uses a supervised method. During the author response period, we add another comparison with a recent paper Sun et al “Amodal Segmentation through Out-of-Task and Out-of-Distribution Generalization with a Bayesian Model” (CVPR 2022) as shown in the below table. Together with the works mentioned above, hopefully we have a reasonable coverage of related work in the later version.
>
> | | Full IoU | Occluded IoU |
> |------------------------|---------------|---------------|
> | Sun et al. (CVPR 2022)  | 0.7837 | 0.1897 |
> | SaVos |  **0.8270** | **0.3159** |
>
> We will also clarify that self-supervision is on the amodal mask completion, where we don’t need human labelers to label the occluded part. There are indeed discussions on whether to use self-supervised or weakly supervised in this setting. In Zhan et al. (CVPR 2020), they pretrained an UNet to get modal segmentation supervisedly and the title for their paper is “Self-Supervised Scene De-occlusion”. While in Ling et al. (NeurIPS 2020), they are using the same setting but referring it as weakly supervised. We will emphasize self-supervision is only for the amodal mask completion in the later version.
>
> **2. Ablation between different architectures and different formulations.**
>
> Thanks very much for this suggestion. We add an ablation experiment that uses SaVos architecture but the loss described in the Self-Deocclusion Zhan et al. (CVPR 2020) paper. The results are shown as follows:
>
> | Architecture | Loss | FISHBOWL Full IoU | FISHBOWL Occluded IoU | KINS-Video-Car Full IoU | KINS-Video-Car Occluded IOU |
> |------------------|------------------|--------------|-----------------------|--------------------|-----------------------------|
> | Self-Deocclusion | Self-Deocclusion | 0.8678 | 0.6418 | 0.8158 | 0.1790 |
> | SaVos | Self-Deocclusion | 0.8742 | 0.6826 | 0.8121 | 0.2040 |
> | SaVos | SaVos | **0.8843** | **0.7111** | **0.8270** | **0.3159** |
>
> When both training with the self-supervision method in Zhan et al. “Self-Deocclusion” (CVPR 2020), SaVos architecture introduces around 0.03-0.04 performance gain. After replacing the loss into SaVos loss, the model achieves another 0.04 performance gain on FISHBOWL and 0.11 gain on KINS-Video-Car. This table disentangles the contribution of architecture and self-supervised formulation.

---

> > ### Author Response · Authors · 2022-08-02
> > **Response to Reviewer CKtM Part 2**
> >
> > **3. Utilize a sota (supervised) method to estimate flow and instance segmentation to see how the accuracy on the flow and segmentation can influence the amodal performance. Potentially also try unsupervised instance segmentation.**
> >
> > Thanks very much for this suggestion. It is worth mentioning that the inputs (including
> > all the instance labels (track ID), optical flow and visible segmentation mask) for the KINS-Video-Car are predicted from the network, rather than using the ground-truth. Specifically, as described in L240, “We use PointTrack [40] to extract visible masks and object tracks to drive our video-based alg”. PointTrack (https://github.com/detectRecog/PointTrack) is a SOTA method for multiple object tracking and segmentation. And for optical flow, we use FlowNet2 (https://github.com/NVIDIA/flownet2-pytorch). With predicted modal mask, tracking and optical flow, SaVos is still able to get visually reasonable predictions as shown in the video “kins_car_test_video_vis.mp4” in the supplementary material. And SaVos outperforms the baseline using the same predicted modal mask input. These might be indicators that SaVos is robust on the noise introduced by segmentation and tracking or flow network. On the other hand, the segmentation and tracking GT is not available for every frame in the KINS dataset. We may not be able to run SaVos on GT input for KINS-Video-Car.
> >
> > It’s a great suggestion to also try unsupervised instance segmentation and flow network. Though unsupervised instance segmentation and tracking still remains quite challenging even for most recent works (e.g. SAVi++ , published in June 2022). Meanwhile, we will mark this effort of using unsupervised segmentation and tracking as a future work and will pay attention to the development of related fields such as object-centric learning   to combine new ideas with SaVos.
> >
> > **4.Why is the supervised baseline called “supervised oracle”?**
> >
> > Considering our method does not use the label on the occluded area, we regard the supervised method using those labels as “oracle”. This might be redundant since we’ve explicitly mentioned “supervised”. We will remove the term in the later version if this is confusing. Thanks for pointing this out.
> >
> > **5.Limitation and social impacts can be expanded.**
> >
> > Thanks very much for the suggestion. Specifically, we can expand limitations with some interesting cases in the supplementary “kins_car_test_video_vis.mp4”. For example, the car marked in red from sec 4-6. While the ego vehicle is moving forward, the mask shapes of the same car at the different frames are different. This deformation can be learned in the current SaVos to some extent, but does not work well in this case. This leads to our suggested future works on “Inductive bias from 3D modeling can be introduced to handle more complex ego and object motions.”
> >
> > It also makes a lot of sense to discuss the trustworthiness of the predictions. This leads to the suggested future work on “Variational method can be introduced to handle uncertainty on the occluded area.”
> >
> > We will describe in more detail in the later version.

---

> > > ### Comment · Reviewer_CKtM · 2022-08-08
> > > **Thank you for the response**
> > >
> > > I have read the other reviews and all author responses. The responses contain many new experiments and ablations that make the paper stronger. I am happy to raise my rating to accept.

---

> ### Author Response · Authors · 2022-08-08
> **Looking forward to your further feedback (Reviewer CKtM)**
>
> Thanks again for the detailed review as well as the suggestions for improvement. We summarize our response as below:
> 1. We discuss the related works suggested by the reviewer and discuss the terms use of “self-supervised” or “weakly supervised”.
> 2. We add an ablation experiment between different architectures and different loss formulations to disentangle the performance gain.
> 3. As we have already used a sota model to predict flow and segmentation on the experiment of KINS-Video-Car, we share our experience on how SaVos outperforms the baseline using the same predicted input and gets visually reasonable amodal prediction, both might indicate SaVos can work with the noise introduced by flow and segmentation prediction.
> 4. We discuss our plan of expanding limitations as the reviewer suggested, using the frames in the supplementary video “kins_car_test_video_vis.mp4” and how the findings lead to future work proposals.
>
> We sincerely look forward to your further feedback to see if the response addresses your concerns. Thanks!

---

### Official Review · Reviewer_cqVs · 2022-07-12

**Rating:** 6
**Confidence:** 4
**Soundness:** 3 good
**Presentation:** 3 good
**Contribution:** 3 good

**Summary:**

In this work, the authors mainly consider the amodal segmentation task for moving object in a video. hey propose a self- supervision strategy and a test-time adaptation pipeline for it. It achieves SOTA results on the FISHBOWL and KINS-Video-Car benchmark.

**Questions:**

My main concern is could the authors ease the strong assumption if would still continue in this direction. Maybe the authors could provide the total number of the cases under the strong assumption. Or could provide one option for the rest cases?

**Limitations:**

The paper misses on some important baseline and comparisons in Structured generative model space, that do AMODAL RGB prediction and segmentation on videos, such as PARTS Kabra et al, Scalor (https://arxiv.org/abs/1910.02384), Flow Capsules (https://arxiv.org/abs/2011.13920) and SQAIR (https://arxiv.org/abs/1806.01794).

**Strengths And Weaknesses:**

This paper is well written and easy to follow. The method's novelty is based on the intuition that the occluded part of an object can be explained away if that part is visible in other frames. The core idea makes sense and is novel in some, even in most cases. But according to the Theoretical Analysis on The Loss Design of SaVos in the supplementary material, ‘… with the following assumptions, we claim that the region covered, … 3) each pixel of this object is visible in at least one frame. …’ the assumption might be too strong and it is easy to point out the opposite examples, such as Fig. 3 and the video in the supplementary material.

---

> ### Author Response · Authors · 2022-08-02
> **Response to Reviewer cqVs Part 1**
>
> We thank the reviewer for the detailed review as well as the suggestions for improvement. We are also preparing the new version of the paper based on the suggestions and additional experiments. Our responses to the reviewer’s comments are below.
>
> **1.Too strong assumption in the Theoretical Analysis in the supplementary material. The authors could provide the total number of the cases under the strong assumption. Or could provide one option for the rest cases?**
>
> Thanks. Our analysis shows that SaVos also learns type prior to handle the cases that break the strong assumptions. Learning type prior is an advantage of our learning based method, though it’s harder to pose theoretical analysis compared with spatiotemporal prior. We organize the analysis as follows: we first describe the intuition and inductive bias about type prior learning, then quantitatively verifies that SaVos still works on the cases that break the strong assumption, we then check the consistency between the feature embedding clustering pattern and the object type to verify the learned type prior.
>
> Specifically, as described in Supplementary L48-51 “We emphasize that having global optima of the proposed loss is a necessary but not sufficient condition for aligning the correct amodal segmentation.” Our method can still work well even if this assumption is not exactly satisfied. We did some analysis and discussions in main paper L174-181 and in supplementary Section D L121-152, “our method can still work at least as good as image-level methods since it is also able to capture type prior for known types with the architecture design in 3.3”, “By receiving amodal supervision signals on different parts from different scenes, as long as the model learns that they are amodal signals for the same type, those signals will be accumulated to form the type prior.”, “From the perspective of the architecture inductive bias, the encoder-decoder architecture in the Amodal Maks Completor contains an information bottleneck. This makes it easy to squeeze out type information since type is a concise representation.”
>
> We verify the above statement empirically in Supplementary Table 2 “Quantitatively, we split KINS-Video-Car into two parts. One presumably contains cases that break the “pixel-wise visible in video” assumption. Specifically, we collect **2899 out of 4644 cases** of which the visible masks are touching with other objects in all tracked frames. This criterion roughly picks out the desired cases (roughly 70% of the selected cases break the assumption), for example, parked cars that line up along the roadside with corners invisible. In Supplementary Table 2 (we also copy the table to show below), the Mean-IOU on the occluded part for this subset is not significantly lower than the numbers for the whole set and the other subset. This validates SaVos has the capacity of handling more general amodal scenarios.” This addresses the suggestion by the reviewer to provide the number of cases under the strong assumption and the provide solution for the rest cases. Figure 1 in supplementary or the attached video “kins_car_test_video_vis.mp4” 18-23 sec visually supports the statement “The car marked in blue behind the  closest-in-path-vehicle (relative to ego) has the bottom left part always invisible. SaVos still completed the full amodal mask of the whole car. This is an indicator that SaVos not only just learn spatiotemporal prior, but also type prior.”
> |                           | Full set | Subset 1 (break the assumption) | Subset 2 |
> |---------------------------|----------|---------------------------------|----------|
> | Occluded-IOU         | 0.3159   | 0.3104                          | 0.3250   |
>
> To verify our network learns type prior as well, we added a new experiment in the author response period. We run tSNE on the spatiotemporal embedding for the test samples in FISHBOWL and noticed the data shows a clear clustering pattern. Though objects for the same type might be split into more than one clusters, data points from different classes usually don’t get entangled. Please check the visualization through this anonymous link: [https://ibb.co/wQGpfnj](https://ibb.co/wQGpfnj)

---

> > ### Author Response · Authors · 2022-08-02
> > **Response to Reviewer cqVs Part 2**
> >
> > **2. Comparisons in Structured generative models like Parts, Scalor, Flow Capsules and SQAIR.**
> >
> > Thanks for the suggestions. We have cited and discussed those four papers in the Related works section L84-89. We’d like to further discuss the relation and the differences of those works with SaVos. Generally speaking, we believe that these works are orthogonal and potentially useful to SaVos. For example, their part modeling can be taken as pre-processing step to SaVos to handle more challenging scenarios. Critically, SaVos solves amodal segmentation by spatiotemporal and type priors with specific inductive biases to learn them. In contrast, Flow Capsules is not trained on the whole video to collect spatiotemporal prior and SCALOR does not explicitly model dense correspondence. We found that both are important to achieve high pixelwise accuracy on complex scenes. These models attempt to maximize the likelihood of the whole video sequences during training so as to learn a more consistent object representation and the amodal representation. However, one major challenge for those models is object discovery and representation, and they are tested on simpler datasets. While for SaVos, we focus to develop a method for amodal mask completion on real-world scenarios.
> >
> > Since generative structured models also reported to acquire amodal representation, we also want to check the model performance. We ran the open-source code of SCALOR (https://github.com/JindongJiang/SCALOR) on KINS-Video-Car dataset and discussed the findings in supplementary L191-194: we found out object discovery on this dataset is still too challenging for existing methods. Please have a check on the frame reconstruction visualization through this anonymous link: [https://ibb.co/HrqZLg4](https://ibb.co/HrqZLg4)
> >
> > In the above figure, the reconstructed background also includes pixels of the foreground car. This indicates that the model cannot clearly decompose the scene into background and foreground objects. Without object discovery, no proper amodal prediction can be expected.
> >
> > Meanwhile, we will pay attention to the progress of this related fields to seek for chances to combine with SaVos in future works.
> >
> > **3.  The authors mainly consider the amodal segmentation task for moving object in a video.**
> >
> > SaVos also works for static objects if the camera has ego motion. This covers some use scenarios of autonomous driving or embodied robotics. As shown in the supplementary video “kins_car_test_video_vis.mp4”. SaVos also completes the masks for the static cars parked alongside the road. We will emphasize the use scenarios in the later version.

---

> ### Author Response · Authors · 2022-08-08
> **Looking forward to your further feedback (Reviewer cqVs)**
>
> Thanks again for the detailed review as well as the suggestions for improvement. We summarize our response as below:
> 1. We summarize our analysis about how SaVos handles the cases that break the strong assumptions by learning **type prior**. The analysis ranges from descriptions of the intuition and inductive bias, quantitative results as suggested, visualization evidence, and further verification of the statement based on the intermediate representation clustering pattern.
> 2. We expand our existing discussion about the structured generative models in the related work section of the current version of paper, and also share our experience of running SCALOR code on the challenging KINS-Video-Car dataset. The potential future work is discussed.
>
> We sincerely look forward to your further feedback to see if the response addresses your concerns. Thanks!

---

### Official Review · Reviewer_k3zX · 2022-07-12

**Rating:** 7
**Confidence:** 3
**Soundness:** 3 good
**Presentation:** 4 excellent
**Contribution:** 3 good

**Summary:**

**Problem**: This paper addresses the problem on predicting amodal segmentation masks of objects using full sequences of videos depicting the object in different states of occlusion.

**Solution**: In this work, a novel model is proposed to address this problem which is trained in a self-supervised manner. For each frame, the proposed model predicts an amodal mask and an estimated motion of the object. These outputs are used jointly to estimate the mask in the subsequent frame. The paper proposes a novel objective that provides supervision using the estimated mask on the subsequent frame and the known modal mask on the subsequent frame.




**Questions:**

Some design choices of the proposed method are not entirely explained or intuitive:
- As explained in Line 182-185, usage of the LSTM based model leads to the "cold start problem" *i.e.* amodal predictions on the first few frames are going to be poor due to lack of prior video data. While a workaround has been proposed, wouldn't a more intuitive choice be a transformer-like architecture or even a bidirectional LSTM?
- Why is $\delta V$ necessary? It seems like $V$ and $I$ for each frame should be sufficient to estimate the amodal masks. It would be interesting to see this ablation.
- In addition to the previous question, is the image patch also really necessary to perform this inference? In theory, it should be possible to complete masks given long enough/diverse enough sequences of masks only.

**Limitations:**

The limitations have not been thoroughly discussed. It would be useful to see at least a few examples of failure cases.
The societal impact also has not been discussed.

**Strengths And Weaknesses:**

## Strengths

### Writing
The text of the paper is very well organize and clearly written. The motivation and inspiration for the work have been explained in great detail. To the best of my knowledge, the related work section thoroughly covers the diversity of relevant literature and discusses the relationship to the proposed work. The details of the proposed model and training methodology has been concisely explained.

### Novelty
The architecture of the proposed model is very intuitive. A CNN is used to encode the modal mask, motion and image patch in each frame. An LSTM is used to capture the spatio-temporal relationships. And finally, an auto-encoder is used to predict the amodel mask using the modal mask and the LSTM features. To the best of my knowledge, this is a novel architecture designed specifically for the task of amodal object segmentation.

One of the key novel contributions of this work is the self-supervised objective that takes advantage of modal masks as supervision to estimate amodal masks. I believe this contribution might be of interest to researchers in this domain and possibly could be extended to other domains.

### Results

The results demonstrate that the proposed model can outperform existing work and produce better amodal segmentations. The experimental evaluation is very thorough and covers multiple datasets for this task. Furthermore, the paper presents results using test-time adaptation to perform inference on samples outside of the training distribution. These results demonstrate that the proposed method is well-suited for test-time adaptation and demonstrates improved generalization compared to existing work.


## Weaknesses

Overall there are no major weaknesses of the proposed approach. However, some of the design choices are not entirely explained. I have listed these questions in the Questions section below.

---

> ### Author Response · Authors · 2022-08-02
> **Response to Reviewer k3zX**
>
> We thank the reviewer for the detailed review as well as the suggestions for improvement. We are preparing the new version based on your suggestions and adding new experiments. Our responses to the reviewer’s comments are below.
>
> **1.A more intuitive choice would be a transformer-like architecture or even a bidirectional LSTM to resolve the cold start problem.**
>
> Thanks very much for the suggestion. Transformer is definitely suitable for the temporal modeling here. We can also use the attention weight of transformer to get more intuitions on how the temporal information is helping on the completion. We will mark this as future steps in later version of the paper.
>
> We use bi-directional prediction as mentioned in L182, which resolves the “cold start” issue. Theoretically, this is very close to use bidirectional LSTM. We use two LSTMs instead of one bidirectional LSTM, since we also want the pipeline to work in the online setting, where future frames are not available and we can only use the past frames. Two single-directed LSTMs allows us to drop the backward one. As discussed in the supplementary material Table 1, SaVos still produces reasonable results when removing bi-directional prediction, beating the Self-Deocclusion baseline in Zhan et al. (CVPR 2020). Of course, transformer can also run online if we add a causal mask on the attention.
>
> **2. Ablation study on removing flow input $\Delta V$ and image patch $I$. And the discussion on why they are necessary.**
>
> Thanks very much for this suggestion. This is indeed a useful ablation study and we will put that into the later version of the paper. We run the experiment as the reviewer suggested:
> | Dataset        | Model              | Full IoU   | Occluded IoU |
> |----------------|--------------------|------------|--------------|
> | FISHBOWL       | without_flow       | 0.8508     | 0.6188       |
> | FISHBOWL       | without_img        | 0.8673     | 0.6634       |
> | FISHBOWL       | SaVos paper_reuslt | **0.8843** | **0.7111**   |
> | KINS-Video-Car | without_flow       | 0.7922     | 0.1525       |
> | KINS-Video-Car | without_img        | 0.8025     | 0.2172       |
> | KINS-Video-Car | SaVos paper_result | **0.8270** | **0.3159**   |
>
> As shown in the above table, removing either flow input or image patch harms the performance. We analyze the results as follows:
>
> Removing optical flow input:
>
> Removing flow input hurts the performance by a large gap up to more than 0.1 occluded IoU. In SaVos model design, we need to warp the amodal prediction to the subsequent frame to get supervision. The warping function takes amodal flow as input. As described in supplementary section E, we crop out the objects and rescale them to 64x128. This operation may lose the information of object motion. Then in the current setting, without flow, the network cannot infer object motion from the input thus the pipeline does not work properly. If we do not apply crop and scale, but put the full resolution mask and image as input, ideally the network can infer the flow from the sequence but directly providing the flow would avoid some unnecessary heavy-lifting and let the network focus on learning the completion signal.
>
> Removing image patch input:
>
> 1. SaVos still outperforms the baseline in Zhan et al. (CVPR 2020).
> 2. Image patch provides important information about which part of the object is occluded now. The visible mask left the occluded part as background but image patch might contain the pixels of another object. Then the network would know where to complete.
> 3. Image patch can provide some photomatic guidance for the flow completion, making the warping more accurate to collect more accurate supervision signals.
>
> **3. Limitations and social impact**
>
> Thanks for pointing out. We put the limitations and social impacts in the supplementary material section G and H. We will expand and put them into the main doc in a later version. Specifically, as described in supplementary L187, “Inductive bias from 3D modeling can be introduced to handle more complex ego and object motions.” This might help on some failure cases shown in the supplementary “kins_car_test_video_vis.mp4”. While the ego vehicle is moving forward, the mask shapes of the same car at the different frames are different. This deformation can be learned in the current SaVos to some extent, but there are cases that do not work well. For example, the car marked in red from sec 4-6. We can tell the deformation is not properly modeled. We mark introducing 3D modeling as a future work to make the model work better in these cases.

---

> ### Author Response · Authors · 2022-08-08
> **Looking forward to your further feedback (Reviewer k3zX)**
>
> Thanks again for the detailed review as well as the suggestions for improvement. We summarize our response as below:
> 1. We run an ablation study on removing flow input $\Delta V$ and image patch $I$ as suggested. We also provide the discussion on why they are necessary.
> 2. We discuss the motivation of using bi-directional predictions with two separate LSTMs instead of one bidirectional LSTM. And discuss the future work using transformer for temporal modeling.
> 3. We discuss our plan to expand the limitation part using the frames in the supplementary video “kins_car_test_video_vis.mp4”, basically how failure cases lead to our proposed future work.
>
> We sincerely look forward to your further feedback to see if the response addresses your concerns. Thanks!

---

### Meta-Review · Area_Chair_t8ju · 2022-08-28

**Recommendation:** Accept
**Confidence:** Certain

**Metareview:**

The paper develops a system for amodal object segmentation in video, trained in a self-supervised manner by requiring consistency between amodal and modal masks, as well as amodal masks estimated for a frame and those temporally propagated according to estimated object motion.  After rebuttal and discussion, all reviewers favor accepting the paper, citing novelty of the approach and convincing results.  The Area Chair agrees with the reviewer consensus.

**Award:**

No

---

### Decision · Program_Chairs · 2022-09-14

Accept